# IL-1β-mediated adaptive reprogramming of endogenous human cardiac fibroblasts to cells with immune features during fibrotic remodeling

Jamila H. Siamwala [1,2✉], Francesco S. Pagano [1], Patrycja M. Dubielecka [3], Malina J. Ivey[4], Jose Pedro Guirao-Abad[4], Alexander Zhao[1], Sonja Chen [2,5], Haley Granston[1], Jae Yun Jeong[1], Sharon Rounds[2,6], Onur Kanisicak[4], Sakthivel Sadayappan [7] & Richard J. Gilbert[8]

The source and roles of fibroblasts and T-cells during maladaptive remodeling and myocardial fibrosis in the setting of pulmonary arterial hypertension (PAH) have been long debated. We demonstrate, using single-cell mass cytometry, a subpopulation of endogenous human cardiac fibroblasts expressing increased levels of CD4, a helper T-cell marker, in addition to myofibroblast markers distributed in human fibrotic RV tissue, interstitial and perivascular lesions in SUGEN/Hypoxia (SuHx) rats, and fibroblasts labeled with pdgfrα CreERt2/+ in R26R-tdTomato mice. Recombinant IL-1β increases IL-1R, CCR2 receptor expression, modifies the secretome, and differentiates cardiac fibroblasts to form CD68-positive cell clusters. IL-1β also activates stemness markers, such as NANOG and SOX2, and genes involved in dedifferentiation, lymphoid cell function and metabolic reprogramming. IL-1β induction of lineage traced primary mouse cardiac fibroblasts causes these cells to lose their fibroblast identity and acquire an immune phenotype. Our results identify IL-1β induced immune-competency in human cardiac fibroblasts and suggest that fibroblast secretome modulation may constitute a therapeutic approach to PAH and other diseases typified by inflammation and fibrotic remodeling.

[1] Department of Molecular Pharmacology, Physiology and Biotechnology, Brown University, Providence, RI, USA. [2] Warren Alpert Medical School of Brown University, Providence VA Medical Center, Providence, RI, USA. [3] Division of Hematology/Oncology, Department of Medicine, Rhode Island Hospital, Warren Alpert Medical School of Brown University, Providence, RI, USA. [4] Department of Pathology & Laboratory Medicine, College of Medicine, University of Cincinnati, Cincinnati, OH, USA. [5] Department of Pathology & Laboratory Medicine, Rhode Island Hospital, Providence, RI, USA. [6] Division of Pulmonary, Critical Care, and Sleep Medicine, Department of Medicine, Warren Alpert Medical School of Brown University, Providence, RI, USA. [7] Heart, Lung and Vascular Institute, Division of Cardiovascular Health and Disease, Department of Internal Medicine, College of Medicine, University of Cincinnati, Cincinnati, OH, USA. [8] Ocean State Research Institute, Providence VA Medical Center, Providence, RI, USA. ✉email: jamilasiamwala@gmail.com

Reactive and reparative fibrosis are universal mechanisms that support the preservation and regeneration of vital organs, such as the heart. In the case of Pulmonary Arterial Hypertension (PAH), excess fibrosis in the contractile chambers of the heart leads to maladaptive remodeling, myocardial stiffness, and increased mortality. PAH is characterized by a mean pulmonary artery (PA) pressure ≥25 mmHg, pulmonary vascular remodeling, increased right ventricular (RV) afterload, perivascular and interstitial fibrosis, right ventricular (RV) hypertrophy, and pump failure[1]. Therapies preventing maladaptive fibrotic remodeling have shown favorable outcomes in rat models of PAH[2], although human trials have been inconclusive[2] due, in part, to a lack of understanding regarding the fundamental mechanisms that underlie cardiac inflammation and fibrosis.

Resident cardiac fibroblasts constitute the largest number of cells in the heart (~70%) and are responsible for homeostatic extracellular matrix (ECM) maintenance and conductivity among cardiomyocytes. Activated myofibroblasts assume an epigenetically altered, pro-inflammatory phenotype, secreting cytokines and recruiting macrophages that drive perivascular inflammation[3]. In association with the initiation of the inflammatory state, resident cardiac myofibroblasts differentiate into myofibroblasts, expressing smooth muscle actin (αSMA) and secreting collagen, fibronectin, and other extracellular matrix molecules[4,5]. In addition, cardiac fibroblasts can be reprogrammed into cardiomyocyte-like cells using exogenous transcription factors and miRNAs, indicating that they possess the capacity to switch phenotypes in response to environmental perturbants[6,7]. During myocardial injury, activated cardiac fibroblasts expand and remodel the ECM, leading to interstitial and perivascular fibrosis, increased RV wall thickness, arrhythmogenesis, reduced contractility, and RV failure[8,9]. The pro-inflammatory triggers of cardiac fibroblast activation, pro-fibrotic responses, and the general mechanisms underlying the conversion of the myocardium from an adaptive to a maladaptive state are largely unknown.

Resident myofibroblasts may be activated or recruited in response to damage-associated molecular patterns (DAMPS) originating in the affected myocytes in order to accelerate extracellular deposition of ECM and promote healing[10]. Yano et al. suggested that resident cardiac fibroblasts, but not bone marrow-derived cells, add to the myofibroblast pool in the setting of myocardial repair[11]. The basis for the surge in the numbers of myofibroblasts that are present following myocardial injury is unknown, and the exact role of resident cardiac fibroblasts in myofibroblast accumulation remains unspecified[12]. Recent experiments using periostin and TCF-21 mouse models have traced the lineage of the expanded fibroblast cell populations that result from myocardial infarction[13,14] and have shown that circulating bone marrow-derived cells, CD14+ fibrocytes, smooth muscle cells, and endothelial cells have the capacity to transform into cardiac myofibroblasts through endothelial to mesenchymal transitioning, and may constitute a source of myofibroblasts and a mechanism for ECM deposition[4,15–17]. Lymphocytes, granulocytes, and macrophages, in particular, accrue in the right ventricle (RV) in response to acute-pressure overload injury due to pulmonary artery banding in dogs[18], along with the elevation of various cytokines. Among the cytokines reported to be elevated in PAH, interleukins are believed to promote myocardial fibrosis[19] and enhance chemotaxis and adhesion molecule expression in response to injury in rats[20,21], piglets[22] and humans[23]. Despite the identification of high levels of IL-1β and other cytokines and chemokines in human RV, the pathophysiological role of IL-1β in mediating RV remodeling and fibrosis is poorly understood. While it was previously thought that this cytokine originates exclusively in bone marrow-derived macrophages, it has more recently been shown that activated cardiac fibroblasts express IL-1β[24], suggesting that resident and non-resident cells participate in inflammation resulting from injury and work concertedly to promote regeneration.

In the current study, we used mass cytometry-based single-cell screening, RNA sequencing, and profiling of secreted cytokines, chemokines, proteins, and metabolites to identify a population of human cardiac fibroblasts characterized by features of both mesenchymal and lymphoid origin cells. We show that this population resides in interstitial and perivascular regions of fibrotic RV tissue from patients diagnosed with PAH and in the perivascular and interstitial fibrotic regions in rats treated with SUGEN/Hypoxia[25,26]. We recreated in vitro, a pro-inflammatory milieu using recombinant IL-1β, and showed that human cardiac fibroblasts acquire stemness and form immune marker-positive cell clusters in response to this cytokine. Lastly, we utilized lineage tracing in mice to track the fate of labeled primary fibroblasts upon IL-1β induction, and, thereby, to confirm cellular transdifferentiation of endogenous fibroblasts to cells expressing an immune phenotype.

## Results

### Single-cell multidimensional mass cytometry identifies distinct cardiac fibroblast subpopulation expressing mesenchymal and lymphoid markers

Although several inflammatory subsets have been identified in the myocardium, the role of resident cardiac fibroblasts in the promotion of inflammation remains unclear. Recent studies in human cardiac fibroblasts isolated from patients with heart failure show that these cells are immunocompetent[24]. We sought to characterize pro-immune features of cardiac fibroblasts through immunophenotyping with multiparametric mass cytometry, a method that categorizes cell populations into lineages based on marker expression (Fig. 1a). We also independently established the identity of commercially sourced and validated human primary ventricular cardiac fibroblasts (hVCF), with multiple fibroblast markers αSMA, vimentin, collagen-1, FSP-1, PDGFRβ, periostin and endothelial cell marker, VE-cadherin[27]. While >90% of the cells were positive for all fibroblast markers and negative for the endothelial cell marker, VE-cadherin, <2% PDGFRβ-positive cells were negative for FSP-1, and some FSP-1-positive cells were negative for PDGFRβ (Supplementary Fig. S1a)[28–31]. FSP-1 is also expressed by immune cells and endothelial cells and therefore is not a specific fibroblast marker[32–34]. It has been suggested that all myeloid and lymphoid cells are bone marrow-derived and play a prima facie role in inflammation, repair and tissue regeneration[35]. However, recent evidence suggests that tissue-resident cells including CCR2+ macrophages may, in fact, initiate inflammatory processes[36]. In the setting of myocardial injury, cardiac fibroblasts may epigenetically assume a pro-inflammatory phenotype and secrete cytokines, whereas chemokines may recruit macrophages and lymphocytes before the arrival of the bone marrow immune cells[3]. By staining single-cell suspensions of 3 human male and female donors (all clinical information is provided in Supplementary Table S1) immune lineage markers in the CYTOF panel (Supplementary Table S2), we identified in all human donors a subpopulation of cardiac fibroblasts phenotypically distinct from the central cluster of homogenous cells, which expresses Vimentin, αSMA and CD4, a hallmark T-helper cell marker (Fig. 1b; red dotted lines, Supplementary Fig. S2a–d). Manual gating of the isolated population of cells and re-application of the viSNE algorithm to qualitatively visualize the expression of markers at a single-cell level provided high-resolution maps of the cardiac myofibroblast markers, αSMA and Vimentin (Fig. 1b), along with lymphoid lineage

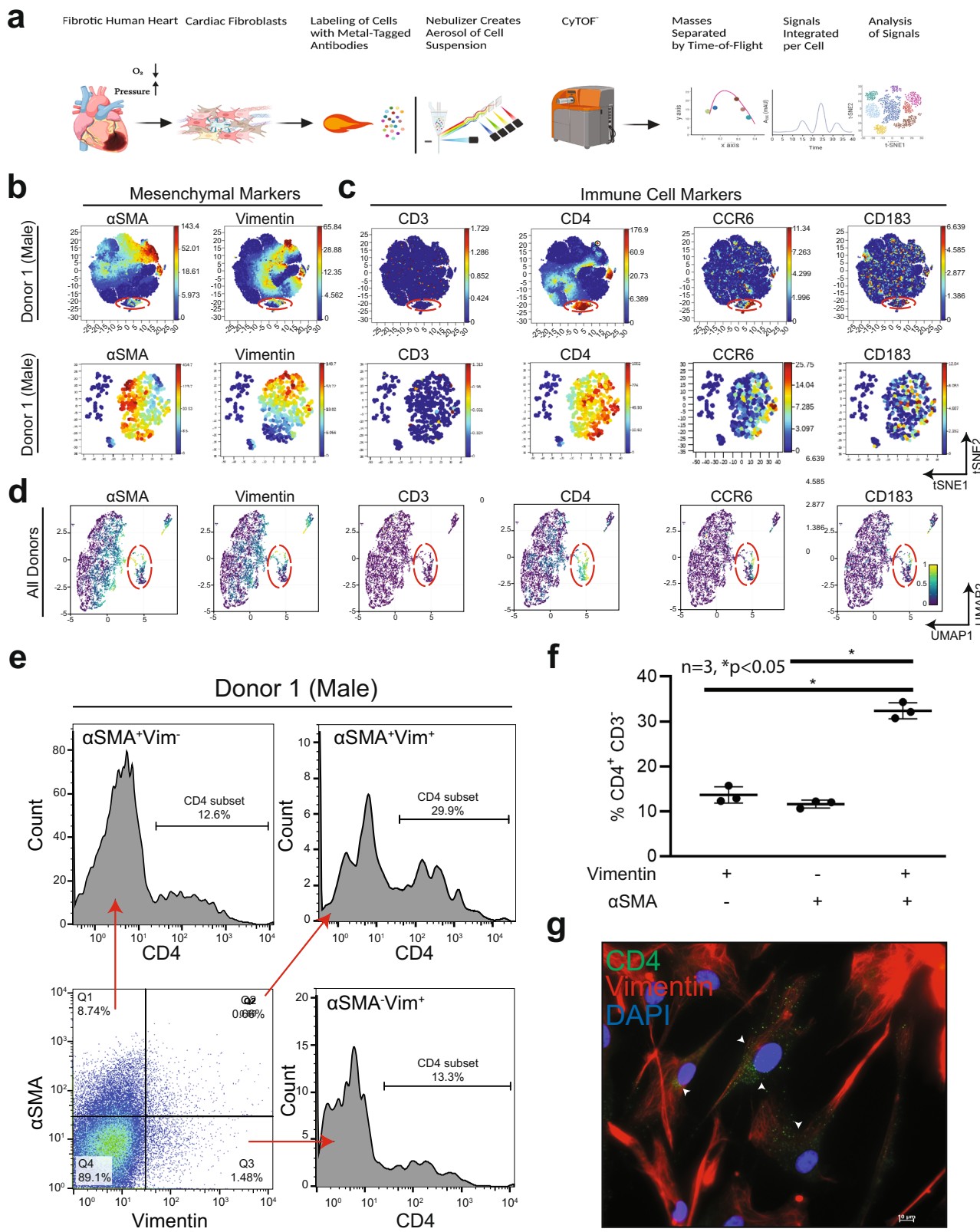

markers, CD4, CCR6, and CD183 (Fig. 1c). Myeloid CD68+ cells (<6%) and HLA-DR+ dendritic cells (<0.8%) were also identified to a lesser extent (Supplementary Fig. S8). Consistent with the viSNE algorithm, a uniform manifold approximation and projection (UMAP) non-linear dimensionality reduction algorithm also showed the same distinct cell population (Fig. 1d). This

approach profiled human cardiac fibroblast cells in isolation as opposed to bulk transcriptomic profiling that can miss smaller populations[37,38]. Measures were taken to ensure that the populations being analyzed were free from beads and only live, nucleated single cells were manually gated, as presented in Supplementary Fig. S1b, c.

**Fig. 1 Identification of a subpopulation of resident hVCFs expressing dual mesenchymal and lymphoid cell surface markers with multidimensional single-cell mass cytometry.** Immunophenotyping of hVCFs derived from human subjects (2 males (ID# 62122, ID# 1281202) and 1 female (ID#534282)) was performed by staining $3 \times 10^6$ cells with heavy metal-tagged antibodies, followed by mass cytometry to identify frequencies of populations in human cardiac fibroblasts. **a** Schematic summary of mass cytometry analysis. hVCFs were immunostained with epitope-specific antibodies conjugated to transition element isotope reporters of different masses. The cells were nebulized into single-cell droplets, followed by the acquisition of an elemental mass spectrum. Conventional flow cytometry methods were used to analyze integrated elemental reporter signals for each cell. This diagram was drawn exclusively by the authors using BioRender (Toronto, Canada) under the aegis of an academic license with Brown University. **b** viSNE graphs of manually gated live, nucleated, hVCF cell clusters expressing varying marker intensities and distribution for a representative donor showed a subpopulation of cells (island) distinct from the majority of cardiac fibroblasts marked by the expression of Vimentin and αSMA+. **c** The hVCFs expressing cluster of differentiation (CD) CD4, CCR6, and CD183 associated with lymphoid cells on hVCF. Protein expression levels are demonstrated on the secondary y-axis scale with blue showing no expression, green, the least expression, and red, the highest expression. Each dot represents the expression profile of a single cell. The red box shows the newly identified population separated from the rest of the cell cluster. Bottom panel of (**b**) and (**c**) are the viSNE of viSNE after gating the newly identified population. **d** UMAP-based clustering of the cells showing the distinct population of cells isolated from the main cluster of homogenous cardiac fibroblast cells. **e** Representative flow cytometry histograms showing frequencies of CD4+ subset of αSMA+Vim+ hVCFs for all the human donor cells. **f** Quantification of CD4+ CD3− cells gated from Vimentin+αSMA−, Vimentin−αSMA+ and Vimentin+ αSMA+ populations is represented as a scatter plot of Mean ± SD ($n = 3$ biological replicates, 2 males and 1 female); *$P < 0.05$, as determined using one-way ANOVA and Tukey's post-hoc multiple comparison. **g** Representative hVCF cells immunostained CD4, Vimentin and DAPI antibodies showing co-expression of Vimentin and CD4. Scale bars are 20 μm.

Quantification of the cells showed that 29.6% of the total cells were Vimentin+αSMA+CD4+, 11.91% ± 2.32% cells of the total cells were Vimentin+αSMA−CD4+ resident cardiac fibroblasts, and 10.53% ± 2% of the total cells were αSMA+Vimentin−CD4+ activated cardiac fibroblasts (Fig. 1e, f) indicating that this cell population expresses different levels of αSMA or Vimentin or CD4 or their combinations. The data for the remaining donors are presented in Supplementary Figs. S4–S7. Finally, we validated the expression of CD4 in Vimentin+ primary human cardiac fibroblast cells by immunostaining and found a punctate staining pattern of CD4 in one or two Vimentin-positive cells (Fig. 1g). To eliminate ambiguity associated with staining artifacts, CD4 antibody specificity was tested by immunostaining of rat spleen sections and blood samples using flow cytometry (Supplementary Fig. S3c). To exclude the possibility of contaminating T-cells, the cells were cultured in fibroblast growth media for at least two to three passages. Besides size-based exclusion, strongly adherent cardiac fibroblasts were washed several times to ensure that any loosely adhered immune cells were removed. Based on these data, we report the presence of a resident primary human ventricular cardiac fibroblast subpopulation that co-expresses mesenchymal vimentin+αSMA+ and the helper T-cell marker, CD4.

**Distribution and expression of spindle-shaped αSMA+ CD4+-co-expressing cells in the fibrotic right ventricle (RV) of patients with pulmonary arterial hypertension (PAH).** Following the identification of a CD4+ human ventricular cardiac fibroblast (hVCF) subpopulation in vitro using mass cytometry, we determined the clinical relevance of this population by mapping the distribution and expression of αSMA+CD4+ cells in autopsy specimens of patients diagnosed with RV fibrosis and PAH. The clinical history and autopsy diagnosis of all the donors designated as either the PAH group or the no PAH group are presented in Supplementary Table S4. Global tissue analysis with Hematoxylin and Eosin (H&E) staining showed variable amounts of cardiac myocyte hypertrophy and age-associated fibrosis in interstitial, perivascular, and subendocardial locations in all donors (Fig. 2a). Sirus red staining identified specific collagen-rich regions and bands of interstitial and subendocardial fibrosis surrounding the myocytes (Fig. 2a). The posterior papillary muscle of the RV showed interstitial fibrosis typical of fibrotic remodeling. The endocardial surfaces of the RV showed hypertrophic changes with big "boxcar" nuclei, hypertrophic cardiac myocytes, subendocardial and interstitial fibrosis, with entrapped cardiac myocytes in the areas of fibrosis (Fig. 2a). To determine

the localization of αSMA+CD4+ cells in the diseased RV, we co-stained formalin-fixed tissues using human-specific αSMA and CD4 antibodies. We noted αSMA cells with a spindle-shaped morphology and membranous CD4 staining in the perivascular connective tissue in cases with minimal interstitial fibrosis and in the dense fibrous tissue but not in non-fibrous tissue (Fig. 2a). Moreover, we found significant increases in spindle-shaped cardiac fibroblasts expressing both αSMA and CD4 in the fibrotic regions compared to the nonfibrotic regions of preserved autopsy RV tissue in humans with a clinical diagnosis of RV hypertrophy and dilation, compared to RVs from those without an autopsy diagnosis of PAH (Fig. 2b). Cells with a round shape and membranous CD4 staining typical of traditional T-cells were predominantly identified inside capillaries (Fig. 2b). Human autopsy tissue is relevant for understanding fibrotic features that phylogenetically distant rodent models fail to fully replicate[39]. We generally observed a higher frequency of cell populations expressing αSMA and CD4 in the human fibrotic tissue compared to tissue obtained from rats and mice having a smaller proportion of cardiac fibroblasts compared to humans.

**Distribution and expression of spindle-shaped αSMA+ CD4+ co-expressing cells in the fibrotic RV in rats treated with SUGEN/hypoxia.** Genetic lineage tracing studies show that resident cardiac fibroblasts derived from a subset of endocardial cells through endoMT mediate pressure overload-induced fibrosis in mice[31]. To determine the role of αSMA+ CD4+ co-expressing cells in myocardial fibrosis, we utilized Fischer rats genetically prone to develop PAH as a model of inducible cardiac fibrosis[40]. Fischer rats were treated with a single intraperitoneal injection of the VEGF inhibitor SUGEN (25 mg/kg) and then subjected to 3 weeks of hypoxia (Hx), followed by 5 weeks of normoxia (Nx) (Fig. 3a). We noted significant increases in the expression of Collagen I/III in both perivascular and interstitial regions in rat RV and LV sections, as determined by Sirius red staining and Masson trichrome staining (Fig. 3b). Control animals were injected with vehicle and housed in room air or Nx for 8 weeks. The SUGEN/Hypoxia rats had lower body weights compared to control rats (Fig. 3c). Fulton index measured from the RV and LV weights using the formula (RV/LV+S) was not significantly increased (Fig. 3d), however, tricuspid annular plane systolic excursion (TAPSE) shows a trend toward reduction in SUGEN/Hypoxia-treated rats (Fig. 3e). None of the other echocardiographic parameters measured, such as cardiac output, ejection fraction, and mean pulmonary arterial pressure, differed

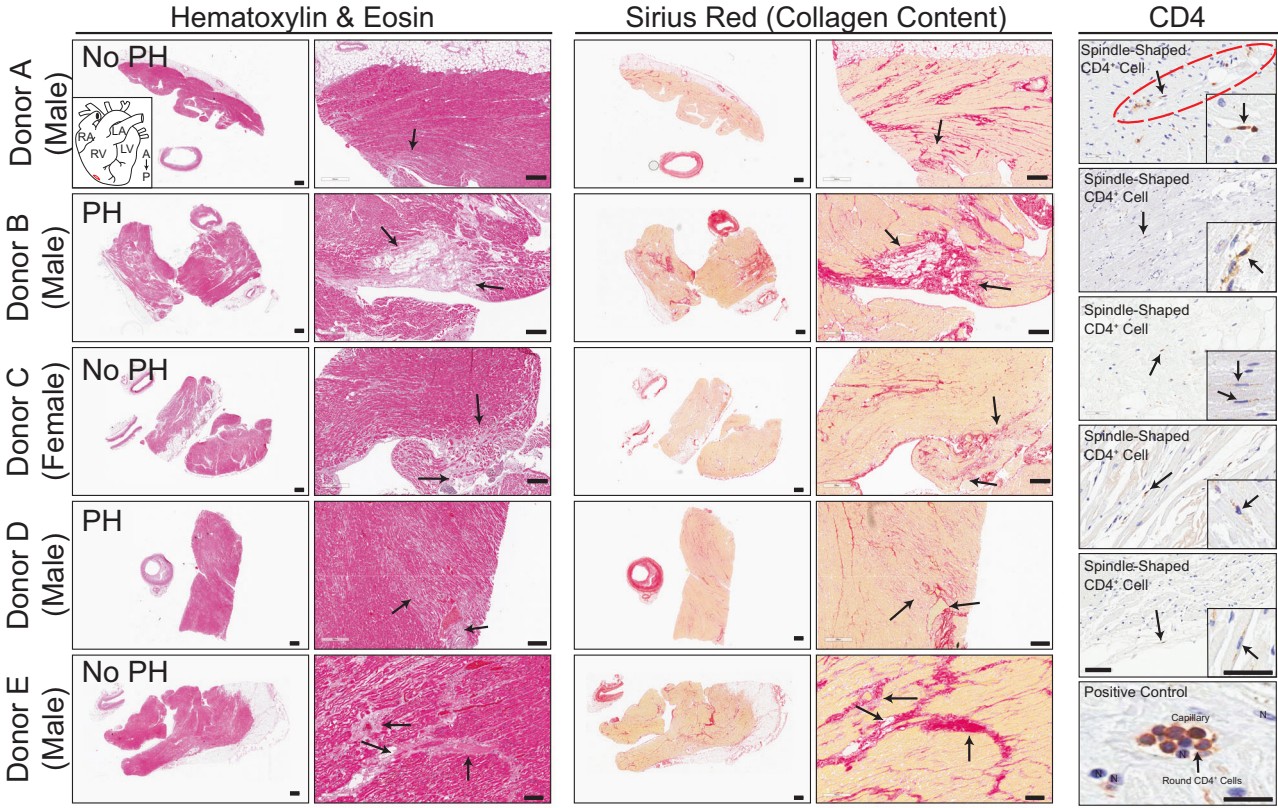

## b Higher Magnification Images of Human Right Ventricle Autopsy Tissue

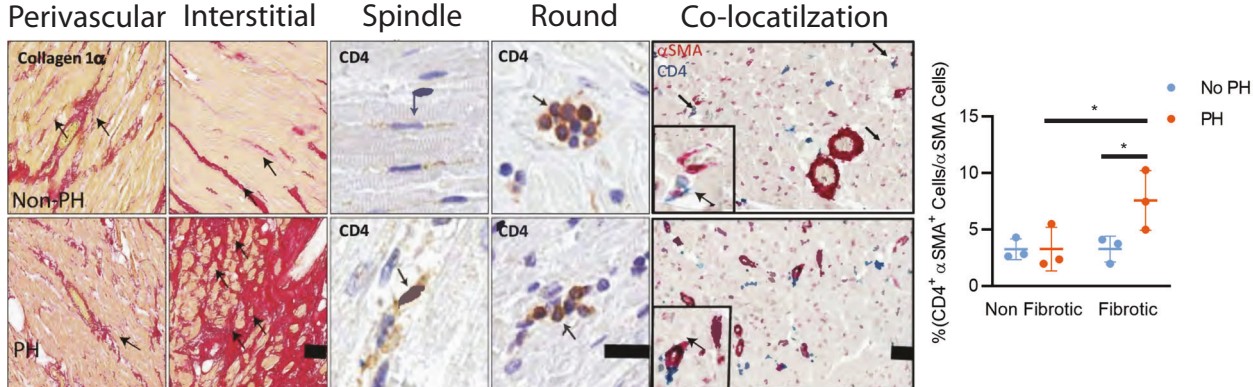

**Fig. 2 Distribution and expression of CD4 in the right ventricle in human cardiac fibrosis. a** Human right ventricular tissues from 5 autopsied donors diagnosed with other conditions (Donor A, C and E) or PH (Donor B, D and F) were stained with standard hematoxylin & eosin (H&E) and Sirius Red. Donors A-E were immunostained for CD4 and counterstained with methyl green to identify nuclei. Donor F was used as a negative control for CD4 immunostaining. The black arrows point to fibrotic areas in H&E-, Sirius Red- and CD4-stained sections. CD4+ expressing spindle-shaped cells in the RV of donor tissue was determined; arrows point to CD4+ spindle-shaped cells. **b** Magnified images of Sirus red staining, CD4 staining (brown) and CD4/αSMA dual-staining in human RV. Arrows indicate the spindle-shaped cells expressing CD4. Quantification of CD4+αSMA+ cells in nonfibrotic and fibrotic regions in the human RV determined by a pathologist blinded to the groups. Scale bars, 75 μm. *P < 0.05, one-way ANOVA and Tukey's multiple comparison test. Quantification of CD4+/αSMA+ manually by a double-blinded observer from at least 15–20 sections/slide for n = 6 samples (*P < 0.05), one-way ANOVA and Tukey's multiple comparison test; Nonfibrotic vs. Fibrotic and no PH vs. PH. Scale bars are 10 μm.

significantly from controls. Further, changes in cardiac fibroblast number and localization in response to SuHx treatment were determined using the fibroblast marker FSP-1 (Supplementary Fig. S10a, b).

To determine the percentage of cardiac fibroblasts expressing CD4 in the right and left ventricle of rat hearts in response to SUGEN/Hypoxia (SuHx), we quantified CD3+ CD4− and CD3+CD4+ primary cardiac fibroblasts populations isolated

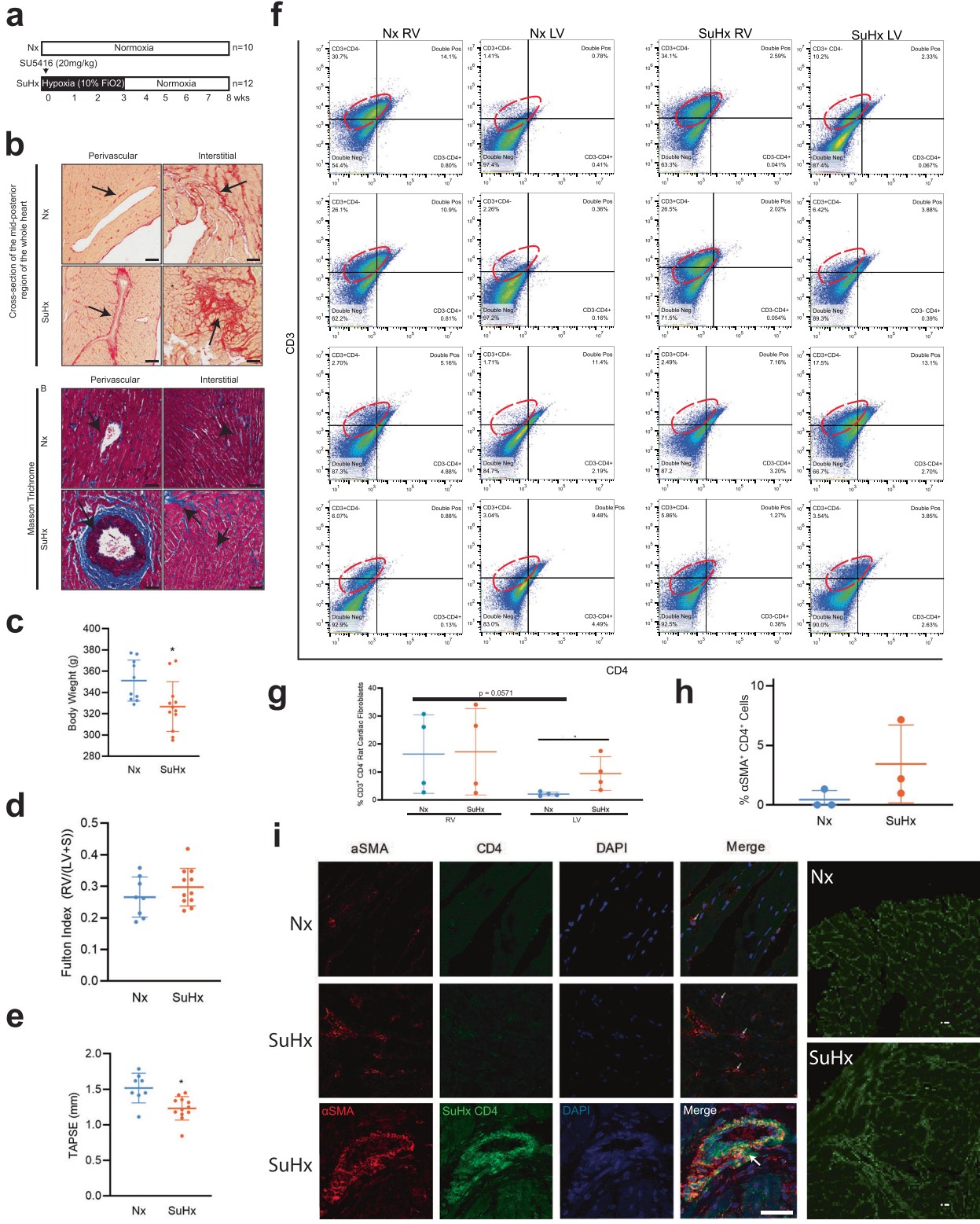

from the RV and LV of Nx and SuHx rats using flow cytometry (Fig. 3f). CD3$^+$CD4$^-$ cardiac fibroblast populations were elevated in RV under both Nx (16.39% ± 14.06%) and SuHx conditions (17.24% ± 15.46%) compared to CD3$^+$CD4$^+$ and were found to be less prominent in LV, under both Nx (2.10% ± 0.71%) and SuHx conditions (9.41% ± 6.04%) (Fig. 3g). In the LV, the percentages of CD3$^+$ CD4$^-$ cardiac fibroblasts were higher in the SuHx conditions and trending toward significance compared to Nx conditions (2.10% ± 0.71% vs 9.41% ± 6.04%, $P = 0.053$).

Percentages of CD3$^+$CD4$^+$ populations expressing cardiac fibroblast cells showed some increase in RV (5.79% ± 4.92%) under SuHx conditions in comparison to Nx, (5.50% ± 5.75%) although it did not reach significance (Fig. 3g). In the RV, there is no change in the percentage of CD3$^+$ CD4$^-$ cardiac fibroblast population after treatment with SuHx. In the LV, there is an increase in the percentage of CD3$^+$ CD4$^-$ cardiac fibroblast population after treatment with SuHx (Fig. 3g). Next, we quantified the number of cardiac fibroblasts co-expressing

**Fig. 3 Histological analysis of the right ventricle in the SUGEN/hypoxia rats. a** Schematic representation of SUGEN/hypoxia for simulating PH in rats. Male Fischer rats were given a single bolus of SUGEN (20 mg/kg) and exposed to hypoxia (10% $FiO_2$) for 3 weeks, followed by exposure to normoxia (Nx) for an additional 5 weeks (SuHx). In parallel, control animals were exposed to Nx (room air) for 8 weeks. The Nx and SuHx groups were comprised of $n = 10$ and $n = 12$ rats, respectively. Quantification of TAPSE trace measured using M-mode echocardiography is presented for Nx and SuHx rats. The Fulton index was determined from the weight of the RV and LV + septum. **b** (Top). Representative images of collagen content in the perivascular or interstitial RV and LV indicated by the Sirius Red-positive regions in the transverse region of Nx and SuHx rats (10 μm thickness), whole heart section cut transversely at the mid-ventral region and perivascular and interstitial regions of both Nx and SuHx animals. (Below) Representative images of Masson trichrome staining the whole heart section and perivascular and interstitial regions of both Nx and SuHx animals. Arrows suggest collagen I-rich regions. Scale bars are 10 μm. **c** Body weight of Nx and SuHx in grams is represented. **d** Fulton Index of Nx and SuHx is represented. Values are mean ± SD ($n = 10$ rats), unpaired $t$-test. **e** TAPSE of Nx and SuHx animals measured using echocardiography. Values shown as the mean ± SD ($n = 10$), unpaired $t$-test. **f** 2D plots depicting cell cytometry of $CD3^-CD4^-$, $CD3^+CD4^-$, $CD3^-CD4^+$, and $CD3^+CD4^+$ in cardiac fibroblasts following induction of hypoxia in the RV and LV of Nx and SuHx animals, with CD3 shown on the vertical axis and CD4 shown on the horizontal axis. The specific $CD3^+CD4^-$ population is encircled by an interrupted red line on each graph. **g** Quantification of $CD3^+CD4^-$ cells in the RV and LV under Nx and SuHx conditions derived from cell cytometry is shown. **h** $αSMA^+CD4^+$ cardiac fibroblast cells following induction of hypoxia protocol in the right and left ventricle of Nx and SuHx animals. **i** High magnification of the perivascular region of Nx, SuHx, and SuHx (x 40) RV tissue expressing $αSMA^+$ (red), $CD4^+$ (green), and DAPI (blue). Collagen I staining shows the collagen expression in the fibrotic regions of Nx and SuHx ventricle (Right panel). Scale bar is 10 μm.

$αSMA^+CD4^+$ cells that accompanied fibrotic changes in the RV of Nx and SuHx rats using confocal microscopy (Fig. 3h). We found three cell populations: T-cells ($αSMA^-$ $CD4^+$), cardiac fibroblasts ($αSMA^+CD4^-$), and cardiac fibroblasts expressing CD4 ($αSMA^+CD4^+$) in rat RV (Fig. 3i). Two distinct morphologies of conventional CD4 cells, round $αSMA^+CD4^+$ and spindle-shaped $αSMA^+CD4^+$ were noted in the RV regions, both which were significantly increased in SuHx compared to Nx rats (Fig. 3i). The number of $αSMA^+CD4^+$ cells tended to be higher in the perivascular fibrotic regions of the RV of SuHx rats compared to control.

**Shifts in human cardiac fibroblast cell population lineages in response to recombinant IL-1β.** Identification of CD4-expressing cardiac fibroblasts in both human and rat RV prompted us to explore possible mechanisms underlying the induction of CD4 expression in cardiac fibroblasts. PAH patients have high levels of plasma IL-1β that correlate with the severity of PAH[41,42], and reduction of IL-1β reduces inflammation and improves right heart function[42]. Based on these and related findings[43–46], we postulated that the pro-inflammatory cytokine, IL-1β, contributes to the induction of CD4 expression in resident cardiac fibroblasts. Our single-cell mass cytometry profiles showed an increase in immunocompetent cardiac fibroblast sub-populations with recombinant IL-1β suggesting a role of IL-1β in cardiac fibroblast re-phenotyping (Supplementary Fig. S8a). Further, CD4 expression varied with IL-1β treatment specifically for male donors (Supplementary Fig. S8b). The non-redundancy scores suggested that CD4 is expressed at levels similar to those of vimentin and αSMA, markers of cardiac fibroblasts (Supplementary Fig. S8c). The percentages of $HLA-DR^+$ dendritic cell, $CD4^+$ lymphocyte, and $CD68^+$ monocyte populations also shifted in response to IL-1β treatment (Supplementary Fig. S8d). $αSMA^+$ and $CD4^+$ expressing cardiac fibroblast populations increased with IL-1β treatment, as seen in the normalized density distribution plots (Supplementary Fig. S8e). The changes in the expression of specific immune cell lineage markers with IL-1β treatment were shown as a heatmap for all three human donors (yellow is high expression and blue is low expression). The vertical pink and green boxes highlighted increases in marker expression in association with IL-1β. Based on the pattern of marker expression ($X$-axis) for all proteins tested, a cell identity can be defined on the $Y$-axis. As the pattern of expression has not previously been described in human cardiac fibroblasts, we have not labeled the lineages on the $Y$-axis[47] (Supplementary Fig. S9a). However, identified the more common patterns for lymphoid lineages, myeloid lineages and memory T-cells and expressed our data in the form of SPADE visualization[48] (Supplementary Fig. S9b–f).

**IL-1β proliferates and differentiates primary human cardiac fibroblasts into immunocompetent cells.** To determine the mechanisms of IL-1β mediated phase shifts in cardiac fibroblast phenotypes to immune cells, we studied IL-1β-mediated cell proliferation and differentiation. Inflammatory processes involving immune cell proliferation, recruitment, reprogramming, secretion of cytokines and chemokines are critical to the IL-1β-linked biological responses[49]. Previous reports suggest that IL-1β is a paracrine growth factor for fibroblasts during intestinal fibrosis[50] and regulates collagenase expression[51]. To determine the proliferative and collagen regulation of IL-1β, hVCFs were treated with vehicle or recombinant IL-1β for 24 h, followed by BrdU and MTT assays and Ki67 staining. IL-1β treatment resulted in an increase in cardiac fibroblast proliferation in a dose-dependent and time-dependent manner (Fig. 4a, b). IL-1β similarly affected the proliferation rates of primary hVCF cells isolated from males and females (Fig. 4c). IL-1β treatment tended to stimulate the deposition of Collagen Iα, as determined from total collagen in the lysates (Fig. 4d) and confocal microscopy (Supplementary Fig. S9g); however, these effects were not statistically different. To determine whether IL-1β differentiates cardiac fibroblasts and induces phenotypic changes, human hVCFs were treated with IL-1β (10 ng/mL) for 96 h, which resulted in the induction of significant morphological changes in hVCF cells by day 4 at this dose and time, including increased detachment and rounding of cells (Fig. 4e). Therefore we used IL-1β dose of 10 ng/mL and 96 h of treatment for all subsequent experiments. Round cells with nonhomogeneous DAPI-stained nuclei appeared with 96 h of IL-1β treatment (Fig. 4e). The round cells, but not the surrounding cells, were positive for both αSMA and CD4 T-cells (Fig. 4e). The round cells were viable and not apoptotic, as determined using trypan blue staining. Additionally, myeloid-specific CD68 positive clusters were also seen in IL-1β hVCF but not in Veh (Fig. 4f). We sought to expand the number of cardiac fibroblasts with T-cell features by growing the cells in T-cell expansion media for 10 days with added IL-1β (Supplementary Fig. S14). We noted the formation of cell clusters in all the human donors, suggesting that IL-1β induced cell clustering from day 7 onward (Fig. 4g). We next characterized the subcellular phenotypic switching of cardiac fibroblasts with IL-1β after 96 h using transmission electron microscopy. Transmission electron microscopy showed more prominent endoplasmic reticulum and Golgi apparatus, suggesting activation and secretory transformation of cardiac fibroblasts in the presence of IL-1β (Fig. 4h). Increased intracellular and budding extracellular microvesicles were seen in the IL-1β-treated cells immunostained with IL-1β receptor (IL-1R) (Fig. 4i). Our data demonstrated that cardiac fibroblasts have

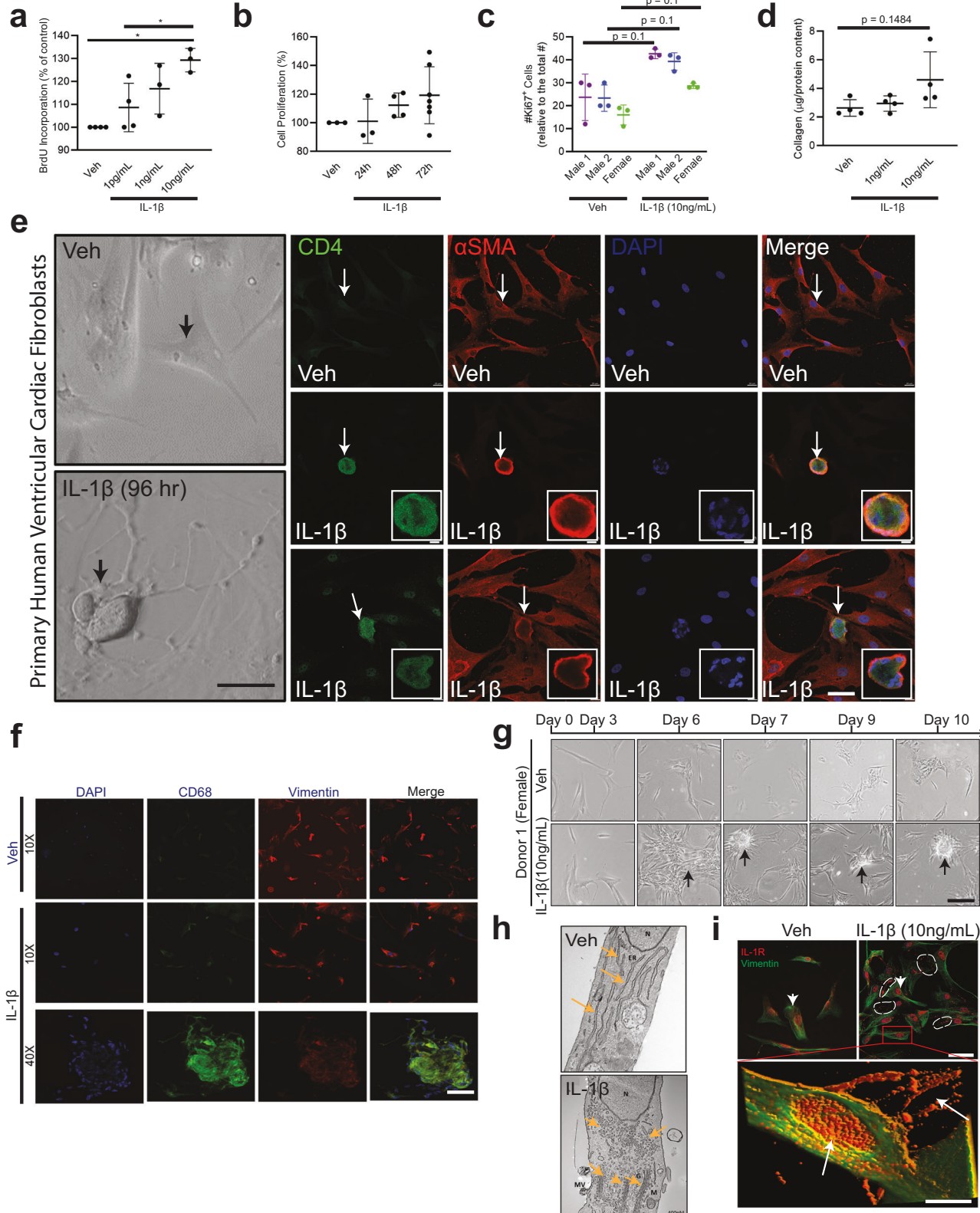

the capacity to assume a secretory cell phenotype in the presence of IL-1β.

**Expression of extracellular matrix and inflammatory genes in response to IL-1β.** Recognizing that differentiated cardiac fibroblasts express both mesenchymal and lymphoidal lineage markers, we evaluated gene responses related to extracellular matrix deposition and inflammatory functions due to IL-1β. We used a targeted qPCR approach gene expression method employing a validated taqman array plate preset with primers specific to extracellular matrix genes or inflammatory genes, specifically assessing the expression of these genes in the presence of Veh or IL-1β. A comprehensive list of both extracellular matrix genes and inflammatory genes modulated by IL-1β is presented as a

**Fig. 4 IL-1β-mediated proliferation, transdifferentiation and activation of primary human ventricular cardiac fibroblasts (hVCF) into CD4-expressing cells in vitro.** Primary hVCF cells (2.0 × 10⁶ cells/mL) from donors (2 males (ID# 62122, ID# 1281202) and 1 female (ID#534282)) were cultured in Fibroblast Basal Media (FBM) for 4 days with 10 ng/mL of IL-1β or in T-cell media with CD3/CD28 T-cell activator for 13 days. **a** hVCF proliferation in response to IL-1β exposure was assessed by BrdU incorporation. Data are Mean ± SD for 3 biological replicates. *$P < 0.05$, one-way ANOVA with multiple comparisons. **b** Cell proliferation response of hVCF to IL-1β after 24 h, 48 h and 72 h of incubation was assessed using MTT assay. Data are Mean ± SD for 3 biological replicates ($n = 3$–6 technical replicates). **c** Cell proliferation response to IL-1β determined by Ki67 staining of nuclei. Data are Mean ± SD for 3–4 biological replicates, $P = 0.1$, one-way Kruskal–Wallis ANOVA with Dunn's multiple comparisons. **d** Collagen content in response to IL-1β doses (1 ng/mL and 10 ng/mL) measured in the cell lysates using the Sircol assay. The data are represented as Mean ± SD values from individual subjects ($n = 4$) biological replicates, no significance (ns), Kruskal–Wallis one-way ANOVA with Dunn's multiple comparisons. **e** Left Panel: Bright-field images of live hVCF cells indicating the shifts in cellular morphology from a spindle-shaped fibroblast to a round cell with a large nucleus in response to 96 h of treatment with IL-1β (10 ng/mL). Arrows indicate a change in morphology of hVCF with IL-1β treatment, but not with vehicle treatment. Right Panel. Immunostaining of fixed hVCF with lymphoid CD4 T-cell marker (green), mesenchymal αSMA (red) and DAPI (blue) markers to characterize the transformed cells. The arrow indicates the emergence of round, multinucleated αSMA⁺ CD4⁺ expressing cells. Representative magnified images of multinucleated notch-shaped giant cells immunostained with αSMA⁺ CD4. **f** Immunostaining of the cardiac fibroblast cell membrane with CD68, Vimentin, DAPI and Merge showing cell clustering phenotype with IL-1B at 40x magnification. **g** Differentiation of cardiac fibroblast in T-cell media with T-cell activators followed for 13 days. hVCF were treated with rIL-1β (10 ng/mL) in fibroblast media for 4 days. Cells were then transferred to T-cell expansion media with T-cell activators (CD3, CD2, CD28) for 10 days. Representative bright-field images taken day 3, day 6, day 7, day 9 and day 10 are presented sequentially. Cells start clustering by day 6 and form prominent circular colonies by day 7. Arrows show multicellular clusters. **h** Transmission electron microscopy micrograph of ultramicroscopic subcellular structures of the vehicle and IL-1β (10 ng/mL) 24 h treated hVCF. N Nucleus, ER Endoplasmic Reticulum, MV Secreted Microvesicle, G Golgi body, M Mitochondria. Scale bar, 400 nm. $n = 3$ technical replicates. **i** Representative immunostained images of Vimentin (green) and IL-1R expression (red) and distribution across the nucleus and cytoplasm of hVCF treated with Veh or IL-1β. The white dotted lines indicate the bright red puncta of IL-1R expression on hVCFs after 24 h treatment with IL-1β. Z stack images of IL-1β treated hVCFs were projected in 3D using image J. Arrows indicate overexpression and extracellular release of IL-1R.

---

heatmap (Supplementary Fig. S11e, f) and in Supplementary Tables S6 and S7. Heatmaps of IL-1β responsive and unresponsive inflammatory and extracellular matrix genes are listed from the highest Z-score to the lowest Z-score (Supplementary Fig. S11a, b). Notably, the CD44 gene, highly expressed during T-cell development, is upregulated 3-fold in IL-1β-treated cardiac fibroblast cells (Supplementary Table S7). The heatmap may reflect the specific transitions of the extracellular matrix and inflammatory gene responses due to IL-1β. Further, relationships between genes, including activation, inhibition, and binding partners of genes differentially expressed with IL-1β (Supplementary Fig. S11c) were represented by STRING analysis (Supplementary Fig. S11g). STRING analysis of significantly expressed inflammatory genes predicts interactions between the IL-1R1 gene and the inflammatory regulator *NFkB1* (Supplementary Fig. S11c, g). In addition, *ANXA1*, which is modulated with IL-1β, was predicted to interact with multiple binding partners, such as *PTAFR*, *LTB4R* and *HRH1*. String analysis of significantly expressed extracellular matrix genes showed a cluster of *ITGA* modulated by IL-1β interacting with binding partners and *CD44* interacting with *MMP14* and *VCAN*. *MMP1* was predicted to interact with *MMP11* and *A2M* (Supplementary Fig. S11c, g). Further, pathway analysis using CYTOSCAPE suggested that immune-responsive genes with a false discovery rate (FDR) of $2.85 × 10^{-13}$ belonged to gene ontology terms "cellular response to cytokine stimulus", "cytokine-mediated signaling pathway", "inflammatory response", "defense response and immune system process" (Supplementary Fig. S11g; Bottom table), while extracellular matrix-associated genes with the FDR of $2.13 × 10^{-17}$ belonged to gene ontology term "extracellular matrix organization" (Supplementary Fig. S11g; Bottom table). We further validated the gene expression of *CCR2* and *IL-1R*, (Fig. 5a) in primary adult rat ventricular fibroblasts isolated using Langendorff's method and expression of *IL-8* and *TGF-β* in primary human ventricular fibroblasts using qPCR (Fig. 5b).

**Induction of immunomodulatory proteins in response to inflammatory cytokine.** In human preadipocytes, IL-1β stimulates the production of cytokines and chemokines in a manner that is similar to the inflammatory response of mature adipocytes[52]. To determine the effect of IL-1β on the cardiac fibroblast secretome (immunomodulatory cytokines, chemokines, proteins, and metabolites), we performed quantitative cytokine and chemokine analysis, proteomics, and metabolomics using MAP Human Cytokine/Chemokine Magnetic Beads and LC-MS/MS from three human donors. Heatmaps of the expression of secreted cytokines and chemokines showed significant changes in the secretome, as represented by orange color (high expression) or blue color (low expression) (Fig. 5c). We noted that 24 h treatment with IL-1β significantly increased secretion of IL-8, IL-10, and MCP-3, while the secretion of IL-6, IL-12p70, TNF-β, and VEGF tended to increase but were not significant. We next optimized a protocol to identify proteins solely secreted in the conditioned media by human cardiac fibroblasts in response to IL-1β using unlabeled proteomics detected by LC-MS/MS. A total of 274 and 301 proteins were secreted in the conditioned media uniquely by cardiac fibroblasts treated with Veh or IL-1β (10 ng/mL) (Supplementary Fig. S12b and Supplementary Tables S8–S11). Out of the annotated 394 proteins, the Venn diagram demonstrated that 120 proteins were unique to IL-1β and that 181 proteins were common to both Veh and IL-1β (Supplementary Fig. S12b). Conditioned media replicates for the Veh and IL-1β groups were internally consistent, as represented in Supplementary Fig. S12c.

Functional gene ontology (GO) enrichment analysis of the conditional media secreted proteins using the g:profiler web server showed humoral immune response, immune effector process, leukocyte-mediated immunity, vesicle-mediated transport, platelet degranulation, neutrophil activation and lymphocyte-mediated immunity as highly significant for a search of biological processes (BP) (Fig. 5e) similar to the gene expression data. In addition, cellular component (CC) enrichment for the secreted proteins was predicted to be a part of the extracellular exosome, collagen-containing extracellular matrix, extracellular matrix, secretory granule lumen and cytoplasmic vesicle lumen (Fig. 5e). In Supplementary Fig. S12d, the gene ontology terms synonymous with functions of secreted proteins are listed, showing the proteins involved in extracellular matrix regulation and inflammation secreted by hVCF in response to IL-1β, as analyzed with PANTHER. Supplementary Tables S6–S11 demonstrate the functions of designated proteins in terms of

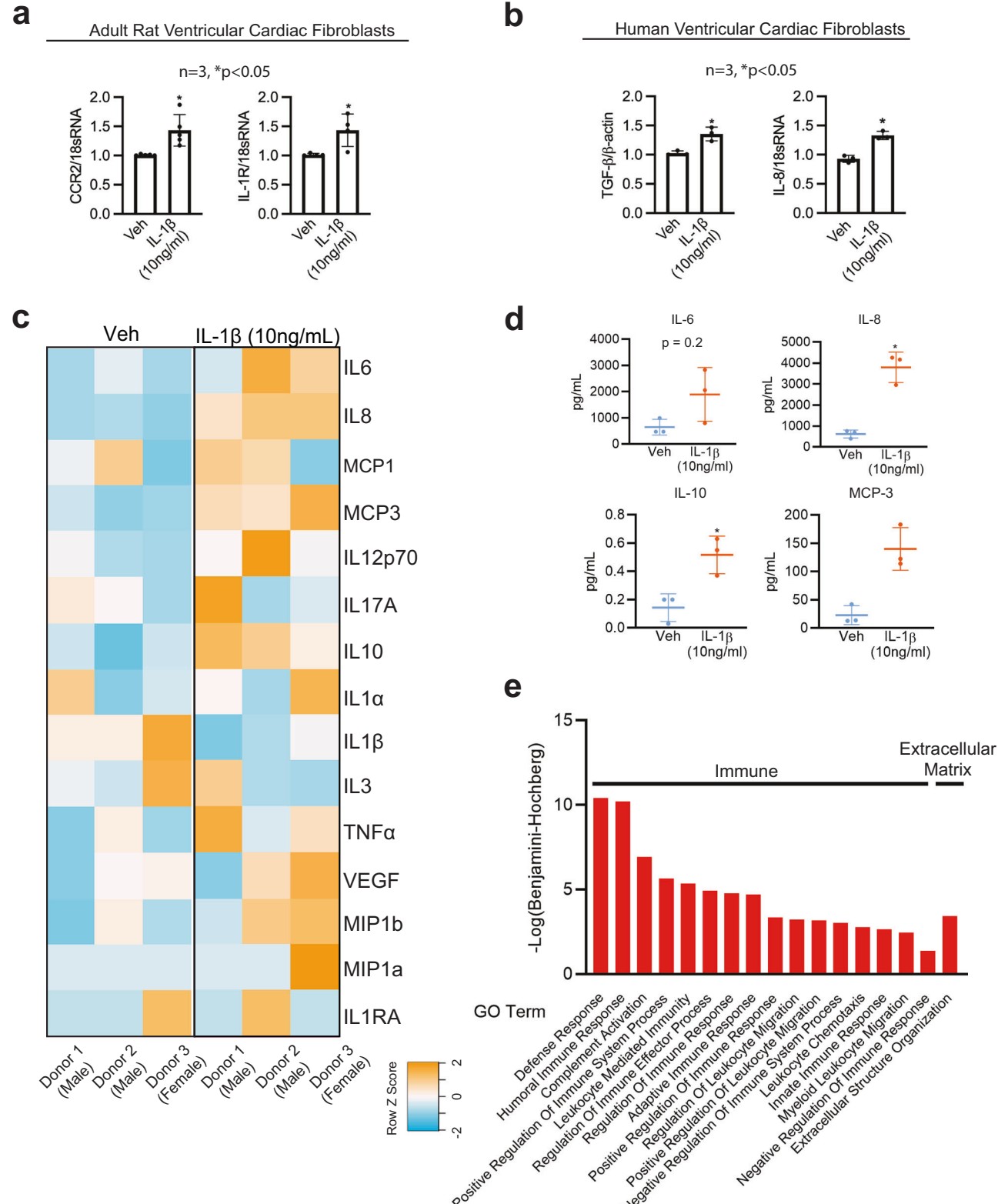

biological processes, molecular function and immune subset population identified uniquely in the conditioned media of IL-1β-treated hVCF. Specifically, proteins involved in the activation of innate and adaptive immune responses, leukocyte activation, and leukocyte migration were displayed in the biological process analysis, molecular function, and immune subset biological processes (Supplementary Fig. S12d–f).

**Transcriptome profile of IL-1β-induced CD4 fibroblast cell population.** Extending our cell cytometry data (Fig. 6a), we characterized the identity of the IL-1β-induced CD4$^+$-fibroblast cell population by flow sorting hVCF cells into CD4$^+$ cardiac fibroblasts and CD4$^-$ cardiac fibroblasts after 10 days treatment in T-cell differentiation media, and then characterized the genetic footprint of the newly identified CD4-expressing cardiac

**Fig. 5 Primary human cardiac fibroblasts secrete cytokines, chemokines, immunomodulatory proteins in response to IL-1β. a** Chemokine CCR2 and cytokine IL-1R gene expression performed in adult rat ventricular cardiac fibroblasts treated with Veh or IL-1β (10 ng/mL) after 24 h using quantitative PCR (qPCR). $n = 3$, *$P < 0.05$. **b** IL-8 and pro-differentiation TGFβ gene expression levels in primary hVCF treated with Veh or IL-1β (10 ng/mL) after 24 h using quantitative PCR (qPCR). $n = 3$, *$P < 0.05$. **c** Heatmap representation of secreted cytokines and chemokines (IL6, IL8, MCP-1, MCP-3, IL12p70, IL17A, IL10, 1L1a, IL1β, IL3, TNFa, VEGF, MIP1b, MIP1a and IL1RA) in the conditioned media after 24 h of treatment with Veh or IL-1β (10 ng/mL) profiled using a milliplex bead-based detection method. The values in the heatmap are expressed in Raw Z-score, ranging from blue (-2) to orange (+2). **d** Scatterplots of cytokines and chemokines (IL-6, IL-8, IL-10 and MCP-3) quantified in the condition media from Veh and IL-1β treated cardiac fibroblasts. The scatterplots represent Mean ± SD values ($n = 3$ biological replicates, *$P < 0.05$) by unpaired-student $t$-test with Mann–Whitney post hoc test. **e** GO Function analysis to identify the function of the 158 secretory proteins unique to conditioned media from IL-1 β treated cells using LC-MS/MS. Benjamini–Hochberg score of false discovery rate.

fibroblasts using next-generation deep total RNA sequencing (Fig. 6b–e and Supplementary Fig. S11b–e). We observed both the unique transcriptomic signatures of the IL-1β induced CD4⁺ hVCF population (Fig. 6b) and differential gene expression between Veh and IL-1β groups (Fig. 6b). Heatmaps based on non-hierarchical clustering showed differences in transcriptomic signatures for the CD4⁺ and CD4⁻ populations (Fig. 6b). The number of significant genes common to all four groups were represented by the volcano plot (Fig. 6c). Venn diagrams showed no common genes shared by the newly identified Veh CD4 and IL1β CD4 populations (Fig. 6d), whereas 255 genes were unique to the IL-1β group (Fig. 6d).

The IL-1β induced CD4 cardiac fibroblast population differed from the parent population by the manifestation of a stem cell-like activated state, as evidenced by the presence of genes associated with pluripotency and reprogramming (*WNT9B*, *SHC4*, *CER1*, *SOX2*, *SHISHA3*, *DLX2*, *SPATA9* and *FOXA2*). Moreover, CD4⁺ stem cell-like cells were secretory, in contrast to the CD4⁻ cells, as evidenced by their expression of microvesicle- and exosome-associated genes (*CCR7*, *CRP*, *ACTA1*, *AZU1*, *CAECAM 8*, *CHMP4C*, *GSTA5*, *CRISP3*, *HIST1H2BC* and *REEP2*). In addition, these stem cell-like secretory cells were metabolically active (*CCR7*, *CRP*, *ACTA1*, *AZU1*, *CEACAM8*, *CHMP4C*, *GSTA5*, *CRISP3*, *HIST1H2BC* and *REEP2*) and possessed characteristics of lymphoidal cells (*FOXP3*, *CD3D*, *IFNL4*, *TMIGD2*, *ILDR2*, *CXCL11*, *ICOS*, *TNFRSF18*, *FCAMR*, *PRKCQ*, *SIT1*, *CCL16* and *NLRP10*), as annotated based on their biological function using Database for Annotation, Visualization, and Integrated Discovery (DAVID) (Fig. 6e). Innate immunity-linked genes (2'-5'-oligoadenylate Synthase Like (*OASL*)), absent in melanoma 2 (*AIM2*) and adaptive immunity-linked genes (*CD1c* molecule, *CD207* molecule), co-stimulatory signal during T-cell activation (inducible T-cell co-stimulatory (*ICOS*)), T-cell chemotaxis (*CXCL11*), TNF receptor superfamily member (*TNFRSF18*) involved in leukocyte migration, B-cell proliferation genes (GRB2-binding adapter protein) and genes associated with phagocytosis (carcinoembryonic antigen-related cell adhesion molecule 4 (*CEACAM4*)) were unique to the IL-1β CD4⁺ group (Fig. 6e). These experiments showed that a subset of terminally differentiated resident cardiac fibroblasts may convert into lymphoid-like cells in the presence of the inflammatory cytokine, IL-1β. Further, we validated the expression IL-1β induced stemness of adult cardiac fibroblast using stem cell-specific markers (*LIN28A*, *NANOG*, *POU5F1*, *SOX2* and *SSEA4*) by immunofluorescence staining. Clusters of differentiated cardiac fibroblast cells expressed all the stemness-associated markers. While *SSEA4* expression was also seen in low quantities in Veh, the expression increased in IL-1β treated cells (Fig. 6f). RNA seq analysis and immunostaining of this unique cell population provided evidence that IL-1β mediates the expression of stemness markers on adult cardiac fibroblasts.

**Inflammatory cytokine (IL-1β)-induced secretion of immune response-associated metabolites.** We analyzed human cardiac

fibroblast-specific metabolites in conditioned media, following treatment with IL-1β, by assaying changes in the metabolic status of cardiac fibroblasts using non-targeted LC-MS/MS (Supplementary Fig. S13). Principal component analysis based on overall differences between the metabolites demonstrated separation of the vehicle group and IL-1β-treated groups (Supplementary Fig. S13a). The volcano plot distinguished metabolites with negative log with a false discovery rate higher than 1 modulated by IL-1β treatment (Supplementary Fig. S13b) Heatmap comparisons of differentially secreted metabolites between vehicle and IL-1β presented in Fig. 6c showed differential metabolite secretion with IL-1β. Interestingly, the metabolic profile for the female donor of the cardiac fibroblast cells with and without IL-1β treatment was different from that of the male donors (Supplementary Fig. S13c). The top 25 differentially upregulated and downregulated metabolites are represented on the heatmap (Supplementary Fig. S13d). Pathway analysis of the differentially expressed metabolites identified significant upregulation of metabolites involved in cardiolipin biosynthesis, a key inner mitochondrial membrane protein and mitochondrial oxidant producer in IL-1β treated cardiac fibroblast cells (Supplementary Fig. S13e). Typical CD4⁺ metabolite signatures of activated T-cells involving methionine metabolic pathways and pyrimidine metabolism, methyl histidine metabolism, phenylacetate metabolism, and de novo triacylglycerol biosynthesis were also upregulated (Supplementary Fig. S13e). Interestingly, metabolites serving as a link between glycolysis and the citric acid cycle, including thiamine pyrophosphate, were differentially secreted in response to IL-1β stimulation. Specifically, metabolites associated with ATP generation, carbohydrate metabolism, and the production of amino acids, nucleic acids, and fatty acids, such as spermidine, spermine and atrolactic acid, were upregulated, while others were downregulated, such as sedoheptulose-1-7-phosphate, oxaloacetate, N6-acetyl-L-lysine, succinate, 1,3 diphosphate glycerate, and citrate-isocitrate (Supplementary Fig. S13f and Supplementary Tables S12 and S13). Our data confirmed that cardiac fibroblasts assumed immune cell features in the presence of IL-1β by upregulating inflammatory and ECM genes and by secreting cytokines, chemokines, immunomodulatory proteins, and metabolites involved in both innate and adaptive immune responses.

**Lineage tracing of cell fate for endogenous cardiac fibroblasts following IL-β induction.** To confirm the cellular identity of endogenous cardiac fibroblasts following IL-1β induction, we utilized an in vivo lineage tracing mouse model. Experimental *pdgfra^CreERt2/+*;R26R-tdTomato mice were generated by combining a fibroblast-specific tamoxifen-dependent cre recombinase-expressing allele knocked into the platelet-derived growth factor receptor-α (*pdgfra*) locus[30,53–55] with a cre-dependent tdTomato fluorescent protein-expressing reporter allele[54] (Fig. 7a). Eight- to ten-week-old mice were fed with chow including tamoxifen in order to promote the permanent labeling of pdgfrα positive cardiac fibroblasts allowing fate change experiments of the pdgfrα

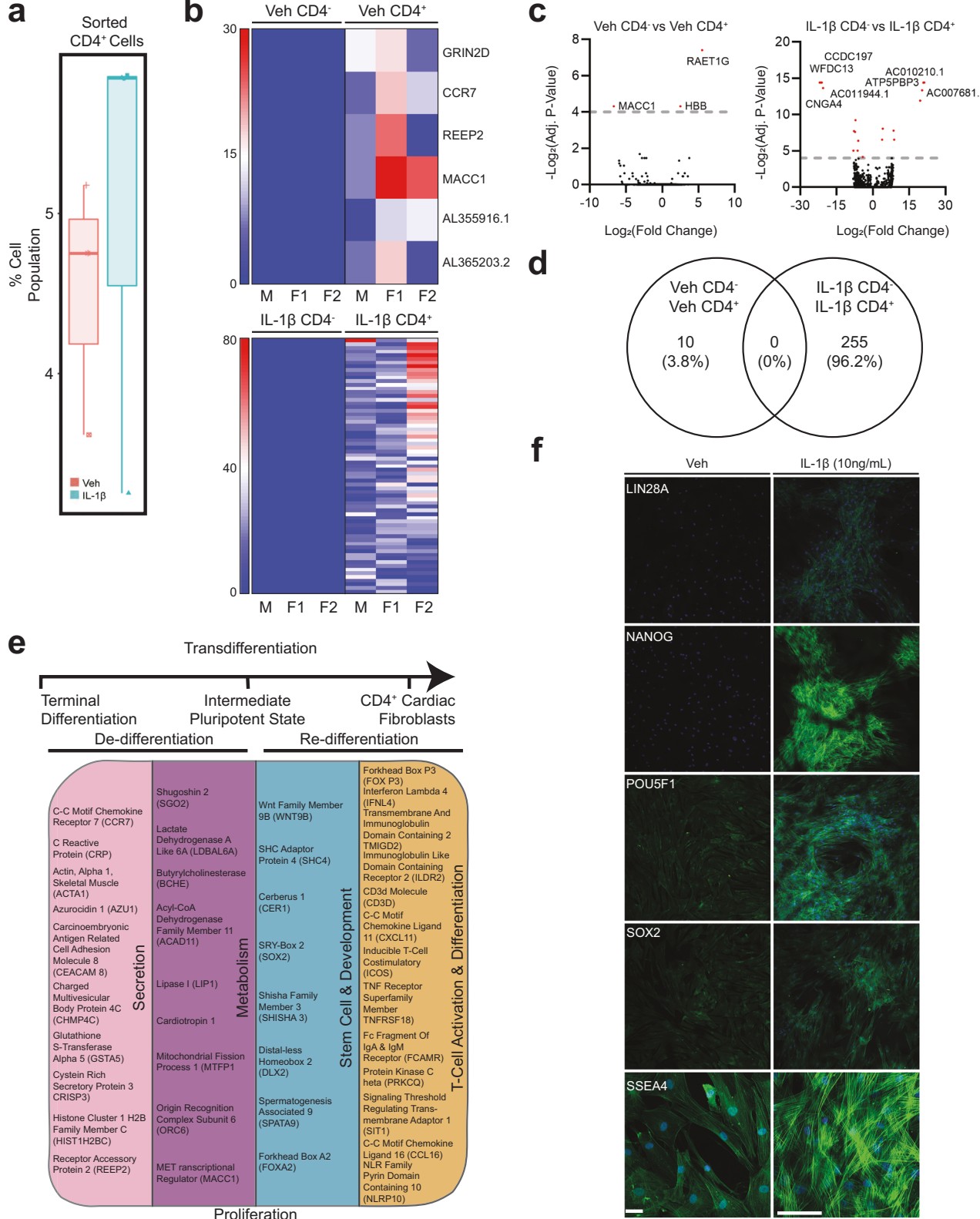

lineage fibroblasts with the reporter allele expression regardless of the *pdgfrα* expression following labeling[54] (Fig. 7b). Utilizing a separate reporter mouse line in which pdgfrα positive cells can be detected with the expression of GFP fluorescence protein driven by the endogenous pdgfrα promoter (*pdgfrα^{GFP/+}*)[53], we demonstrated that the isolated GFP positive cardiac cells are, in fact, pdgfrα positive fibroblasts, which are, prior to induction,

negative for CD4 or CD45. (Supplementary Fig. S14a). Following the in vivo labeling protocol, mice hearts were harvested and enzymatically digested, and then cardiac ventricular fibroblasts were sorted with FACS utilizing the red fluorescence (Fig. 7c). We monitored the stable expression of tdTomato expression within cultured primary cardiac fibroblasts via real-time imaging of the red fluorescence and showed that, regardless of treatment, the

**Fig. 6 Genetic characterization of differentiated CD4⁺cardiac fibroblast cells.** Primary hVCFs from 3 human donors (males and female) were differentiated for 10 days in T-cell media with or without IL-1 β (10 ng/mL) and flow-sorted into CD4⁺ and CD4⁻ populations, followed by next-generation RNA sequencing using the Illumina platform. **a** Box plot of shifts in CD4⁺ human cardiac fibroblast population in response to IL-1 β. **b** Heatmaps showing the expression pattern of genes uniquely identified in Veh CD4⁺, Veh CD4⁻, IL-1 β CD4⁺ and IL-1β CD4⁻ populations. **c** Volcano plot showing the number of significantly upregulated and downregulated genes for each of the comparisons. False discovery rate (FDR) (adjusted *P*-value < 0.01). **d** Venn diagram showing the number of common genes among the Veh and IL-1β among the unique genes identified for all the populations. **e** Summary of gene ontology hits using a database for annotation, visualization, and integrated discovery (DAVID) online platform showing the enrichment of genes involved in secretion, pluripotency, development, metabolism and T-cell activation and differentiation. Benjamini–Hochberg score of false discovery rate. **f** Validation of expression of stemness markers (Lin28A, Nanog, POU5F1, SOX2, SSEA4) in Veh and IL-1β treated cardiac fibroblasts.

reporter allele was expressed at all times (Fig. 7d). Following IL-1β induction, tdTomato cells were subjected to flow cytometric analyses for their expression of fibroblast origin cell identity markers (*mesfk4*[30] and *pdgfrα*) and immune cell markers (CD4 and CD45) (Fig. 7e). Compared to controls, IL-1β treated cells showed a substantial presence of transdifferentiated fibroblasts, which were positive for CD45 and CD4 (Fig. 7e right panels). Upon analysis of the *pdgfrα* lineage traced cardiac fibroblasts that acquired CD4 and CD45 labeling following IL-1β induction, we showed that in 92% of these cells, the fibroblast cell identity markers were lost (Fig. 7f). However, when we analyzed the rest of the cells compared to CD45 and CD4 double positive cells we still detected the fibroblast markers, as shown with overlayed histogram (Fig. 7g). Compared to the CD45 identity, CD4 expression in cardiac fibroblasts seemed to be acquired or detectable much readily even in the vector group, hence we confirmed these results by analysis of the CD4 flow cytometry plots with overlayed histogram (Fig. 7h). Human fibroblasts and mouse cardiac fibroblasts showed a similar conversion rate to immune cell phenotype with IL-1β treatment (Fig. 7i). Similarly, human cardiac fibroblasts that were originally *pdgfrα* positive and CD4 negative were shown to acquire a *pdgfrα* negative CD4 positive phenotype (Supplementary Fig. S14b). We have therefore demonstrated that lineage-traced primary endogenous mouse cardiac fibroblasts lose their fibroblast identity upon IL-1β induction and acquire an immune phenotype.

## Discussion

Myocardial injury is characterized by complex interactions involving mesenchymal and immune cells that collectively form the backbone of regenerative repair in the heart[56,57]. We identify, through an array of methods in tissues, isolated cells, and secreted media, along with in vivo lineage tracing, a subpopulation of hVCF cells expressing surface markers of quiescent cardiac fibroblasts (Vimentin), activated cardiac fibroblasts (αSMA) and lymphoid T-cells (CD4). We specifically confirm the presence of spindle-shaped cells dually expressing αSMA and CD4 in fibrotic right ventricle tissue of humans diagnosed with PAH and right ventricles of rats subjected to SUGEN/hypoxia to model PAH. We demonstrate that these resident cardiac fibroblast cells are structurally and functionally similar to lymphoid cells and that they produce a prototypical fibro-immune secretome (genome, proteome, metabolome) related to both extracellular matrix deposition and inflammation when exposed to the pro-inflammatory cytokine, IL-1β. Characterization of this IL1-β-induced CD4⁺ population using deep RNA sequencing suggests a unique population of genes involved in the stem cell-like dedifferentiation and redifferentiation and metabolic reprogramming of fibroblasts to lymphoid cells. Lastly, cell fate assessment through lineage tracing confirms that IL1-β induction of endogenous cardiac fibroblasts results in the loss of cardiac fibroblast markers and the gain of immune cell markers in 92% of cardiac fibroblasts profiled. These findings are consistent with a novel role for endogenous resident fibroblasts in carrying out a first

response to pressure-induced cardiac injury through immune modulation of fibrosis[43,58] (Fig. 8).

PAH is a syndrome characterized in part by immune dysregulation involving CD4 helper T-cell type I (Th1)/Th17 immunity. Modulating CD4 Treg cells reduces endothelial injury and prevents PH in T-cell-deficient rats[59,60]. Mature CD4-depleted mice are protected from left ventricular fibrosis, but not cardiac hypertrophy in transverse aortic constriction (TAC)-induced left ventricular failure[61]. The mechanisms by which CD4 Treg cells potentiate or alleviate fibrosis are currently unknown. It has been speculated that CD4 Treg cells that are recruited from the bone marrow secrete extracellular matrix similar to cardiac fibroblasts, which promotes thickening and reduces compliance of the ventricular wall and leads to systolic and diastolic dysfunction.

Both CD4 T-cells and cardiac fibroblasts are identified independently in fibrotic regions secreting cytokines, chemokines and extracellular matrix deposition and remodeling suggesting that these are two different cell types. We postulate, based on our findings, that these are functions of single cells that switch phenotypically in response to IL-1β. Our data suggest that pro-inflammatory stimuli prompt endogenous resident cardiac fibroblasts to undergo genetic reprogramming and phenotypic conversion into immune cells, which are then capable of amplifying the inflammatory response and recruiting other immune cells. T-cell macrophage cell fusion is common in the setting of myocardial injury, resulting in the formation of giant cells. We show the appearance of notch-shaped nonhomogeneous giant cells after treatment of cardiac fibroblasts with IL-1β for 4 days. Our RNA-seq data demonstrate that CD4 cardiac fibroblasts exhibit features of mesenchymal cells, T-cells, and pluripotent cells in response to IL-1β treatment. These results suggest that terminally differentiated cardiac fibroblasts may dedifferentiate into a transient stem cell-like state and then re-differentiate into lymphoidal cells in response to IL-1β, thereby proposing new methods for regenerating damaged cells and tissues[62]. The specificity of IL1-β induced fibroblast plasticity was confirmed by lineage tracing demonstrating IL1β-dependent transformation of the endogenous fibroblasts to a phenotype largely devoid of characteristic mesenchymal markers and conversion to a CD4 immune cell phenotype. The epigenetic and metabolic reprogramming seen in the CD4 population is reminiscent of the induction of lymphocyte-dependent adaptive immune responses and trained immunity[63]. The regulatory pathways involved in cardiac fibroblast activation, proliferation, differentiation, and transdifferentiation in response to IL-1β are currently unknown.

The work presented in this manuscript has several limitations. Regulation of cell phenotypic switching in response to tissue perturbations, such as depicted here, is likely to involve multiple cell types and cytokine pathways in addition to IL-1β. While we cannot exclude the involvement of other cytokines and chemokines in cardiac fibroblast reprogramming into lymphoid lineages, our work indicates that IL-1β alone is sufficient to transdifferentiate a subset of the cardiac fibroblast population into CD4 cells, possibly through dedifferentiation into stem cell-like

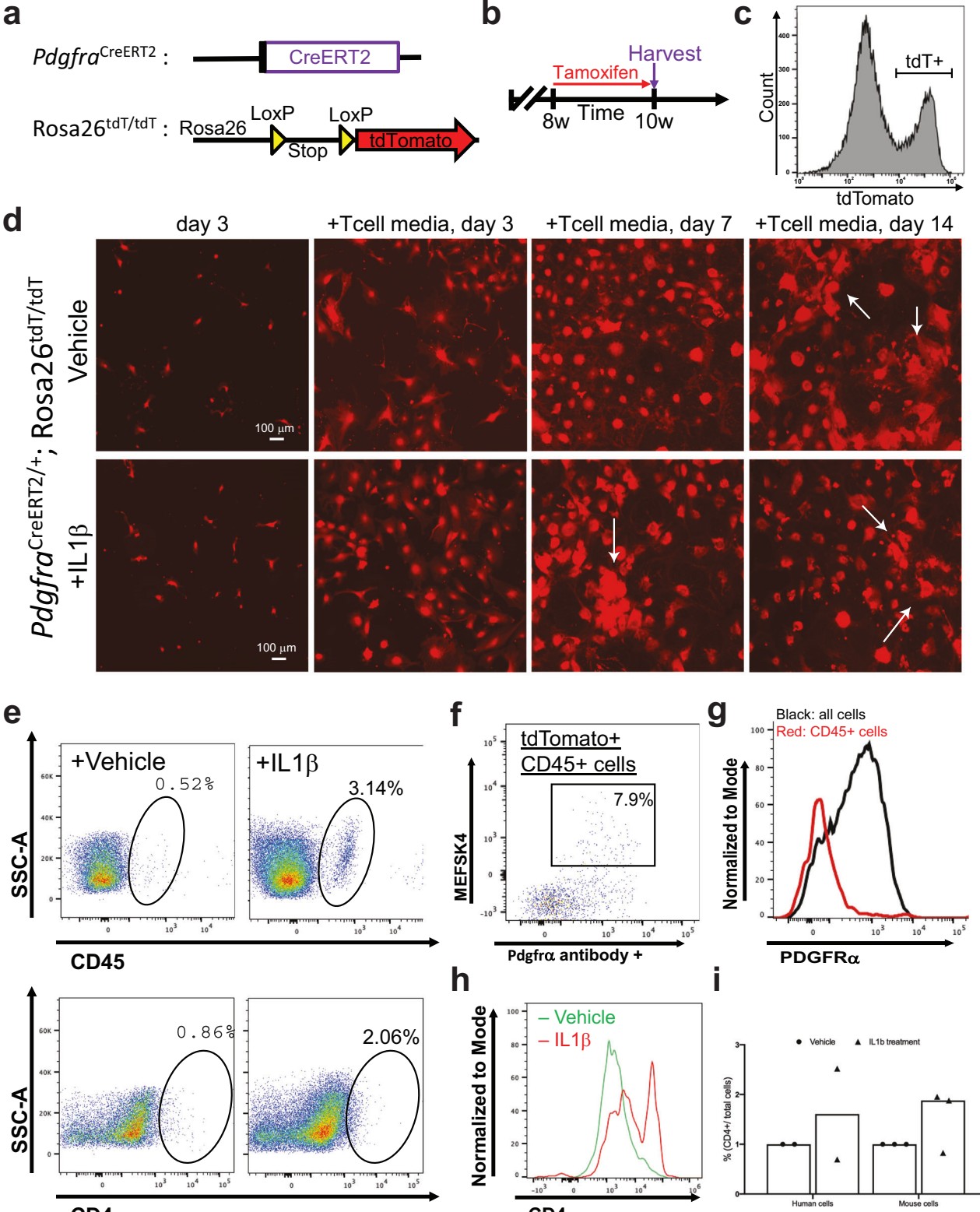

cells and then redifferentiation in a T-cell factor-enriched environment. Moreover, we have not addressed the possibility that reprogramming may, in fact, be bidirectional, with phenotypic switching from cardiac fibroblasts to lymphoid cells coupled with similar switching from lymphoid cells back to fibroblasts. Notwithstanding, we have characterized using complementary in vitro and in vivo methods the presence and biological significance of unique CD4-expressing cells in the adult heart. We note that we were unable to quantify the CD4-expressing cardiac fibroblast in mice with only 20% fibroblast population which may reflect a reduction in leukocyte numbers in general, as previously reported in human pulmonary fibrotic tissue[64].

Recognizing the potential role of fibroblast plasticity in orchestrating the conversion, enlistment, and activation of an

**Fig. 7 Lineage tracing of cell fate for endogenous cardiac fibroblasts following induction via IL-β administration.** Lineage tracing of Pdgfrα+ cardiac fibroblasts was performed to confirm that these explicit cells become CD4-expressing cells in association with IL β1 induction. **a** Generation of the lineage tracing mouse model in which tamoxifen-dependent Cre recombinase is expressed under the influence of a Pdgfra promoter. Combined with a Cre-dependent tdTomato (red fluorescence) expressing reporter, the allele permanently labels all resident cardiac fibroblasts. **b** Experimental scheme in which PdgfraCreERT2/+; Rosa26tdT/tdT mice are given tamoxifen chow for two weeks prior to harvesting to tag cardiac fibroblasts in vivo. **c** Flow cytometric sorting of tdTomato labeled cardiac fibroblasts. Bracket shows the tdTomato positive population that is sorted and cultured. **d** Live red fluorescence imaging of cultured fibroblasts. Throughout all of the steps of the culturing protocol, tdTomato signal has been maintained due to the permanent recombination of the reporter allele. (Scale bar is 100 mm). **e** Flow cytometry analysis showing CD45+ and CD4+ phenotype acquisition of tdTomato+ fibroblasts following IL1β treatment. **f** Flow cytometry analysis demonstrating that originally Pdgfra+ cardiac fibroblasts do not maintain their fibroblast markers upon transitioning to immune phenotype. **g** Normalized histogram analysis of Pdgfra positivity among tdTomato+CD45+ cells. **h** Normalized histogram analysis of CD4 positivity from flow cytometry among tdTomato+ cells. **i** Quantification derived from flow cytometry of both mouse and human fibroblasts that acquired CD4 phenotype, shown as a percentage of CD4 cells/total cells sampled. Each experiment has a biological $n = 3$ and technical replicates of $n = 3$.

array of inflammatory cell types to the stressed myocardium, therapeutic methods designed to modulate the cardiac fibroblast secretome should emerge. Since right ventricular failure is the most common cause of mortality in PAH, the modulation of fibroblast plasticity through physiological perturbation raises the possibility of treatments based on the controlled transition from maladaptive to adaptive remodeling.

## Methods

**Human right ventricle tissue and human cardiac fibroblasts.** Approval for the use of human tissue was obtained from the Institutional Review Board of the Providence VA Medical Center (IRB-2019-038). De-identified transverse right ventricular tissue sections (10 μm thickness) derived from patients with a diagnosis of PAH and from patients who died of other unrelated causes (control) were obtained post-mortem from the Yale Tissue Repository Service, New Haven, Connecticut. Autopsy diagnoses of donors are described in Supplementary Table S4. Human ventricular cardiac fibroblasts (hVCF) were obtained from Lonza (Walkersville, MD) (Cat# CC-2904) with the following lot numbers (Lot# 67771, 62122, 1281202, 534282, TL210281) and a purity of >99%. Characteristics of human cardiac fibroblast donors are described in Supplementary Table S1.

*Animal models.* Animal experiments were approved by the Institutional Animal Care and Use Committees (IACUC) of the Providence VA Medical Center and the University of Cincinnati. The care and use of animals were performed in accordance with guidelines of the Institutional Animal Care and Use Committee of the Providence VA Medical Center (IACUC-2019-005) and the University of Cincinnati (IACUC-21-04-16-01).

*SUGEN/hypoxia rat model of cardiac fibrosis.* Twenty male Fischer (CDF) rats (F344/DuCrl) weighing 220–250 g were obtained from Charles River Laboratories. The animals were divided into two groups: Normoxia (Nx; $n = 10$) and SUGEN/Hypoxia (SuHx; $n = 10$). Rats were administered a single intraperitoneal injection of the VEGFR inhibitor, SUGEN 5416 (Cayman Chemical; 20 mg/kg). The animals in the SUGEN/Hypoxia group were then exposed to 3 weeks of normobaric hypoxia (10% FiO₂, Biospherix, Ltd, Parish, NY), followed by 5 weeks of normoxia[41,65]. The animals in the control group received a single intraperitoneal injection of the vehicle and were exposed to normoxia for the entire duration of the study (8 weeks in total). Fulton's index, a weight ratio of the RV divided by the sum of LV and septum, was measured to determine the extent of RV hypertrophy. Cardiac fibrosis was determined by Sirus red staining and Masson trichrome staining in addition to eosin and hematoxylin staining. The images were analyzed using Orbit

Image Analysis V3.15 software, which can be trained to measure the entire section.

*Fibroblast-specific CreERT2 TdT reporter transgenic mice.* The $pdgfra^{CreERt2/+}$;R26R-tdTomato mice were generated by combining a fibroblast-specific tamoxifen-dependent cre recombinase-expressing allele that is knocked into the platelet-derived growth factor receptor-α (*pdgfra*) locus[53–55] with a cre-dependent tdTomato fluorescent protein-expressing reporter allele. The Pdgfrα-CreERT2: B6.129S Pdgfratm1.1(cre/ERT2)Blh/J were obtained from Jacksons Laboratory (Strain #:032770).

*Echocardiography.* Rats were anesthetized using isoflurane inhalation and subjected to transthoracic echocardiography using a Vevo 2100 with an MS250 transducer at 13–24 MHz at baseline and after 3 weeks of hypoxia to measure RV fractional area and free wall thickness, pulmonary acceleration time (PAT), pulmonary ejection fraction and tricuspid annular plane systolic excursion via a parasternal short-axis view at the mid-papillary level. Pulse wave Doppler echo was used to measure pulmonary blood outflow at the levels of the aortic valve to measure PAT and PET. TAPSE was measured by aligning an M-mode cursor s to obtain an apical four-chamber view and aligning it as close to the apex of the heart as possible. Stroke volume (SV), fractional shortening (FS), ejection fraction (EF), and cardiac output (CO) were measured from the left ventricle (LV). The cardiovascular parameters were analyzed on Vevo Lab 3.2.0 using 3 different heart cycles averaged. Rats were euthanized by an overdose of CO₂, which was continued for 1 min after cessation of breathing.

*Non-Langendorff method of rat cardiac fibroblast isolation and flow cytometry.* Rat cardiac fibroblast cells were isolated from the right and left ventricles and processed for flow cytometry[66]. Upon cutting the rat thoracic cavity to expose the heart, EDTA buffer was injected into the RV through the aorta to freeze contractions. The heart was clamped and removed. EDTA buffer was injected into the apex of the LV. Next, a perfusion buffer was injected into the apex of the LV. Finally, the collagenase buffer was injected into the LV via 3 injections until digestion was apparent. The clamp was removed, and the heart was split into the RV free wall and LV plus septum, respectively, and weighed to measure RV/LV plus septum weight ratio (Fulton index). The RV and LV were separated, manually dissociated, and incubated on tissue culture plates for 20 min at 37 °C to isolate adherent cells. The supernatant containing non-adherent cells was collected and plated on tissue culture plates to ensure maximal collection of adherent cells. The population enriched for non-adherent cells was then collected by centrifugation at 1500 rpm for 5 min and used for downstream analysis. Collected cells were stained with PE-conjugated anti-rat CD3 antibody and FITC conjugated anti-rat

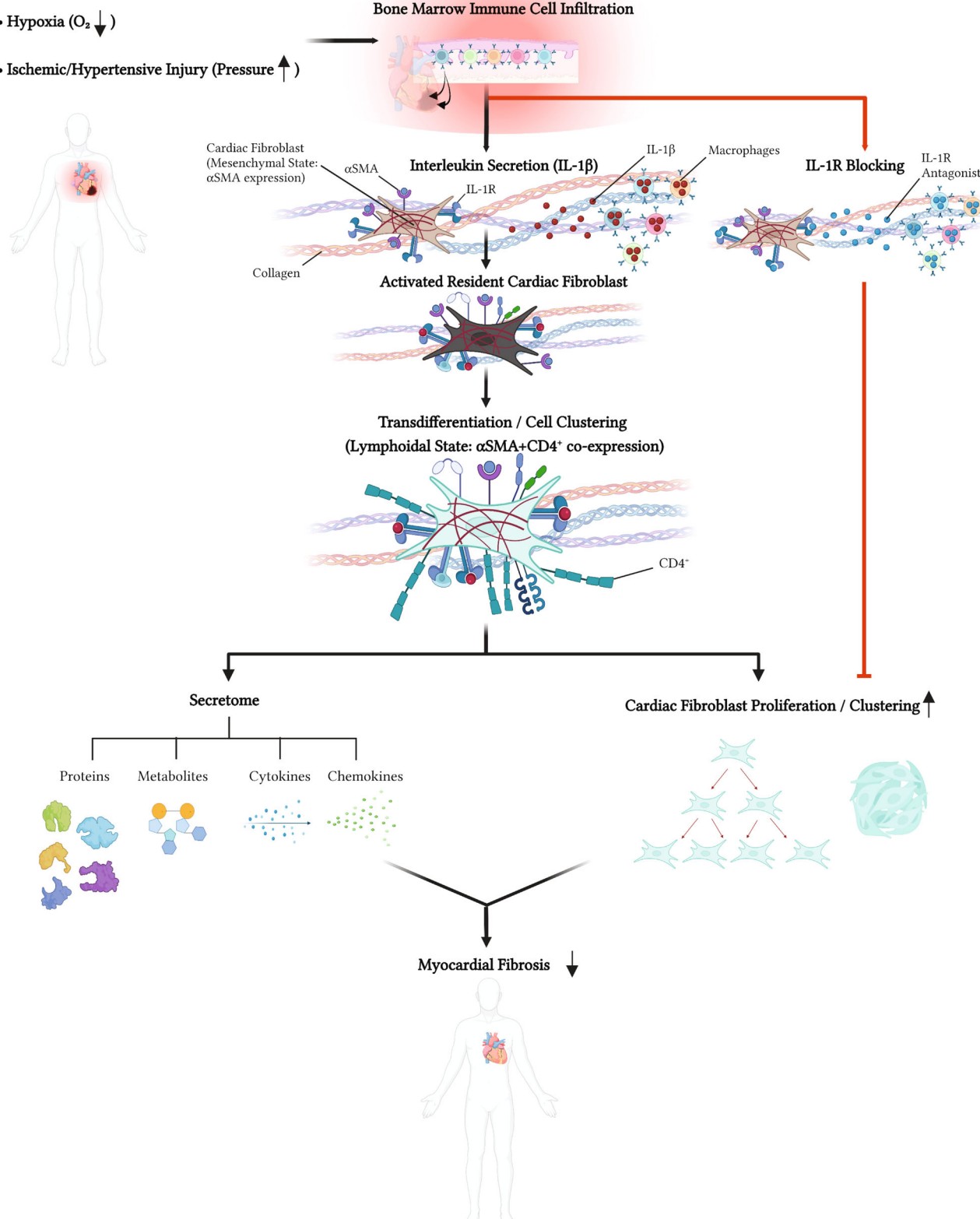

**Fig. 8 Proposed role of endogenous fibroblast transdifferentiation in the pathogenesis of right-sided heart failure due to myocardial fibrosis.** We propose that resident cardiac fibroblasts undergo staged polarization/phenotypic switching in response to pro-inflammatory signals, such as IL-1β, to facilitate tissue repair and cell survival. This results in the transdifferentiation of endogenous fibroblasts to a secretory cell with a lymphoid cell-like phenotype that participates in the amplification and regulation of the inflammatory response through the secretion of immunomodulatory proteins, metabolites, cytokines, and chemokines. Enhancing IL-1β expression may enhance the participation of resident cardiac fibroblasts in the processes of cardiac inflammation and repair. This diagram was drawn exclusively by the authors using BioRender (Toronto, Canada) under the aegis of an academic license with Brown University.

CD4 antibody (Biolegend, San Diego, CA) for 1 h and subjected to flow cytometry using BD FACS AriaIIIu flow cytometer. Data were analyzed using FlowJo software v10.0.6.

*Immunohistology of human right ventricular tissue.* Transverse sections (10 μm thick) of formalin-fixed, paraformaldehyde-embedded (FFPE) tissues from the apex of the right ventricle from individuals diagnosed with PAH or controls were stained with hematoxylin & eosin (H&E). Unstained human right ventricular tissue sections were subjected to immunohistochemistry detecting CD4 (Cat # ab133616, clone EPR6855). The FFPE sections were de-paraffinized in xylene, dehydrated in 100% ethanol, 95% ethanol, 70% ethanol, 50% ethanol and rehydrated in water. Antigen retrieval steps to unmask the antigens were performed using Tris-EDTA buffer (100 mM Tris base, 1 mM EDTA, 0.05% Tween-20, and pH adjusted to 9.0) and steamed at 100 °C for 20 min. The endogenous peroxidase activity was blocked using 0.3% peroxide solution for 30 min. The slides were incubated in 2.5% normal serum for 1 h at room temperature. CD4 primary antibody was used at a concentration of 1:500 and incubated at 4 °C overnight. Slides were washed, incubated with secondary antibody for 30 min, and visualized using 3, 3' diaminobenzidine peroxidase substrate system. Morphometry was performed on both the round and spindle-shaped cells expressing CD4. The nucleus was counterstained with methyl green. Pathological assessment was done in random order on the entire right ventricular tissue sections by a blinded observer. Nuclei were quantified using orbit image analysis software (v3.15). The specificity of the CD4 antibodies was validated on OCT-embedded rat spleen sections by immunostaining.

*Immunohistology of rat ventricular tissue.* Animals were euthanized and the RV and LV dissected and fixed with 30% formalin. The Optimal Cutting Temperature (OCT) blocks with the samples were sectioned at the mid-ventricular heart region (5 μm thickness) by a cryostat before the immunostaining procedure. Samples were then washed in PBS (2x), blocked with 5% normal serum, 1% BSA, and 0.3 M glycine in PBS and then incubated with rabbit monoclonal anti-CD4 and mouse monoclonal anti-αSMA (Supplementary Table S2) followed by incubation with secondary antibodies. The slides were then washed with PBS (3x), mounted with Vectashield Antifade Mount Medium with DAPI, and visualized using a Zeiss LSM 800 Airyscan confocal microscope. Quantification of the CD4 and αSMA regions was performed using orbit image analysis. The individual points represent the average fluorescence intensity of areas captured from 5–10 fields per section for a total of 2 sections. Fibrosis was assessed on a 4% paraformaldehyde-fixed transverse section of the mid-heart region using standard Masson Trichrome and Sirius Red Staining captured using Aperio Scan Scope. The individual data points represented an average of 10–15 positively stained areas.

*Primary human ventricular cardiac fibroblast culture and stimulation.* Human ventricular cardiac fibroblasts (hVCF) were obtained from Lonza (Walkersville, MD) (Cat# CC-2904) with the following lot numbers (Lot# 67771, 62122, 1281202, 534282, TL210281) and a purity of >99%. According to the company records, the cells were isolated from the heart explant during transplantation. Cells were isolated and propagated from heart explants obtained during transplantation for 3 male and 2 female subjects ($n = 5$). The donors were between 63 and 73 years of age and were deemed to be normal. Clinical characteristics of the human subjects are presented in Supplementary Table S1. The hVCF cells (passage 3–8) were seeded at the density of $2 \times 10^5$ cells/75 cm² flasks in medium supplemented with FGM™-3

Fibroblast Growth Medium-3 BulletKit™ (Cat# CC-4526), 0.1% gentamicin/ amphotericin-B (Gibco, Thermo Fisher Scientific, Waltham, MA, USA) and incubated in a 37 °C and 5% $CO_2$ incubator.

*Characterization of primary human ventricular cardiac fibroblast culture.* The hVCF were characterized by morphology, contact inhibition[67], adherence[47] and immunostaining for collagen-I, vimentin, fibroblast-specific protein (FSP-1), platelet-derived growth factor receptor-β (PDGFRβ), α-smooth muscle actin (αSMA) and periostin. The cells were fixed with 4% paraformaldehyde for 10 min at room temperature and washed with PBS. Next, cells were permeabilized with 0.1% Triton-100 and blocked with 5% BSA for 2 h at room temperature[29,68]. The list of antibodies is provided in Supplementary Table S2. The primary antibodies were diluted (1:100) in the blocking buffer and incubated with the cells at 4 °C overnight. The cells were then washed 3x PBS and incubated with species-specific Alexa Fluor 488 conjugated goat anti-rabbit or Alexa Fluor 594 conjugated goat anti-mouse secondary antibodies (1:300; Invitrogen). The cells were washed 3x in PBS and mounted on a 1.5 mm coverslip with a Prolong™ Gold Antifade Mountant with DAPI.

*Mass cytometry analysis.* hVCF ($3 \times 10^6$) cells treated with vehicle or IL-1β (10 ng/mL) were subjected to staining with a mass cytometry panel of 26 metal conjugated antibodies, thereby allowing the identification of myeloid, lymphoid, B cells, natural killer cells and dendritic cells (Supplementary Table S3).

*Panel design.* Our cytometry dataset consisted of 200,000 cells analyzed using 28 validated markers and manually gated cell populations with a total number of 3 million cells. Our panel design was based on the identification of immune functions of cardiac fibroblast cells upon stimulation with IL-1β. We designed the panel to identify the immune cell surface markers and to classify the unidentified cardiac fibroblast population based on prior evidence of immune cell lineages and validated antibodies. Our panel consisted of antibodies that labeled T-cells, NK cells, B cells, monocytes, and dendritic cells[69].

*Cell labeling.* Cell labeling was performed on a CyTOF 1.0 mass cytometer (Fluidigm, San Francisco, CA, USA)[47]. Briefly, the cells were incubated with cisplatin, 25 μM, Enzo Life Sciences, Farmingdale, NY. Then, $3 \times 10^6$ cells were washed with rinsed with 1X PBS and resuspended in 50 μL of 1X PBS and 1% BSA containing metal-tagged antibody cocktail for 30 min at room temperature for the immune cell antibody panel (Supplementary Table S3). The cells were washed twice with PBS and then fixed with 1.6% paraformaldehyde. Cells were washed again with PBS and 1% BSA and incubated overnight at −20 °C. Cells were stained with an iridium DNA intercalator (Fluidigm, San Francisco, CA, USA) for 20 min at room temperature. Cells were washed with PBS and water before re-suspending in 1X EQTM Four Element Calibration Beads (Fluidigm, San Francisco, CA, USA) and collected in a CyTOF 1.0 mass cytometer (Fluidigm, San Francisco, CA, USA). Events were normalized as described previously[47].

*Data processing and analysis.* The fcs files generated by the mass cytometry were pre-gated on viable cisplatin and nucleated cells (iridium+). Single, viable, nucleated cells were selected by gating using CYTOF Clean R command on the fcs files generated from the CYTOF. Data were analyzed using Cytobank 7.3.0 (Santa Clara, CA, USA)[70]. The raw median intensity values corresponding to the expression of the immune lineage markers were transformed to hyperbolic arc sine (arcsinh) with a cofactor of 5 for all the datasets. SPADE trees, viSNE, FLOWSOM[70] and

hierarchical clustering were performed using published algorithms[70]. We used FlowSOM clustering algorithms to annotate the clusters into six major immune lineages on the basis of known lineage markers: (1) CD4 T-cells, (2) B cells, (3) NK cells, (4) dendritic cells, (5) lymphocytes (6) myeloid cells and (7) unidentified population of cells. We used the transformed median expression of each lineage marker for all the cells in the particular cluster to quantify the protein expression[47]. The raw median intensity values corresponding to the expression of the immune lineage markers were transformed to hyperbolic arc sine (arcsinh) with a cofactor of 5 for all the datasets. Next, to specifically examine the CD4 T-cell population in greater detail, we manually gated the highly CD4-expressing T-cell population and produced t-SNE maps for the CD4 T-cell population. We used the median expression of each lineage marker for all the cells in the particular cluster[47]. In addition, we performed the uniform manifold approximation and projection (UMAP), a non-linear dimensionality reduction technique based on a manifold learning technique that preserves local neighborhood relationships and global structure. The code for running UMAP is uploaded as a separate document.

*Cardiac fibroblast culture in ImmunoCult^TM–XF T-Cell Expansion Media.* Cells were cultured in FGM^TM-3 Fibroblast Growth Medium-3 BulletKit^TM (Cat# CC-4526, Lonza, Basel), 0.1% gentamicin/ amphotericin-B (Gibco, Thermo Fisher Scientific, Waltham, MA, USA) for 96 h with or without IL-1β at a concentration of 10 ng/mL before replacing the media with ImmunoCult^TM –XF T-cell expansion media (Cat # 10981, Stem Cells technology, Cambridge, MA) supplemented with IL-1β. ImmunoCult^TM Human CD3/CD28 T-Cell Activator was added after 72 h of culture in ImmunoCult^TM –XF T-cell expansion media. Cells were cultured in T-cell expansion media with T Cell Activator in the presence or absence of IL-1β for 14 days.

*Morphometric analysis of cell clustering.* Morphometry was performed on cell clusters at 100× magnification in a randomized, blinded manner based on layering of cells forming a rosette, weaving or lattice pattern, and formation of a 3D structure visually identified by its white color and boundaries.

*CD4 antibody validation.* Rat blood was aliquoted into FACS tubes and stained with either CD4 PE or CD3 APC or the isotypes diluted in FACS buffer. Samples were vortexed, covered with aluminum foil, and incubated at room temperature on gentle shaking for 45 min. BD Pharm Lyse^TM (2–8 mL) was then added to each sample to lyse the RBC, vortexed, covered with aluminum foil, and incubated for another 15 min on low shaking. Cells were then washed with 2 mL of FACS buffer per sample and centrifuged at 1300 rpm for 10 min at room temperature. The supernatant was decanted and cells were resuspended in 250–500 μL of FACS buffer and covered with aluminum foil until run on BD LSR II flow cytometer.

*TaqMan human extracellular matrix array and inflammation array.* Total RNA was extracted from cardiac fibroblast cells using a Qiagen RNA isolation kit and converted to cDNA using high-capacity reverse transcriptase (Applied Biosystems, Thermo Fisher Scientific^TM, Waltham, MA, USA). Transcripts associated with inflammation and extracellular remodeling were analyzed using the TaqMan^TM Human Inflammation Array 96 well, Applied Biosystems, Thermo Fisher Scientific, Cat #: 4414074, Waltham, MA, USA, and TaqMan^TM Array Human Extracellular Matrix and Adhesion Molecules, 96 well, Applied Biosystems, Thermo Fisher Scientific, Cat #: 4414133, Waltham, MA, USA following the manufacturer's instructions.

*Real-time quantitative polymerase chain reaction (PCR) analysis.* Based on the TaqMan array, the expression level of the selected inflammation and extracellular matrix genes was evaluated using quantitative RT-PCR (qPCR) analysis. Total RNA was isolated from cardiac fibroblasts using RNeasy Mini Kit (Qiagen Inc., Valencia, CA). First-strand cDNA was synthesized from total RNA (10 ng/μL) using a high-capacity cDNA kit (Applied Biosystems) according to the manufacturer's instructions. The SYBR green PCR reactions were performed on the cDNA samples using the SYBR green universal master mix. The fold differences in mRNA expression were calculated by normalization of the cycle threshold [C(t)] value of a target gene to reference gene (α-actin and 18sRNA) using the Livak method. The primers for genes are provided in Supplementary Table S5.

*Transmission electron microscopy.* The hVCF cells seeded on glass coverslips were washed 3x and fixed in 1% glutaraldehyde in 0.05 M cacodylate buffer. Samples were then post-fixed in 1% osmium tetroxide and 1% uranyl acetate. Samples were dehydrated, embedded, and examined using a Philips 410 Transmission Electron Microscope.

*Immunoblotting.* Immunoblotting procedure was performed on primary hVCF cells as previously described by the group[71]. Briefly, $1 \times 10^6$ cells were collected from Petri plates and lysed in radio immunoprecipitation assay buffer containing phosphatase inhibitor cocktail. The tubes were spun at $5000 \times g$ for 5 min and the supernatant was transferred to new tubes. The protein content in the lysate was measured using a BCA protein assay and normalized to yield 100 μg/mL. 1x Laemelli sample buffer (containing 0.1% β-mercaptoethanol) was added and the samples were heated for 10 min at 90 °C. Protein samples were separated based on molecular weight by SDS-PAGE and transferred to polyvinylidene fluoride membranes followed by blocking in 5% Milk or 5% BSA for 1 h. The membranes were incubated with primary antibodies (Supplementary Table S2) in a blocking buffer overnight. The membranes were then washed 3 times with TBST and incubated with secondary antibodies for 2 h in a blocking buffer. The phosphorylated proteins were detected by primary antibodies, which were recognized by horseradish peroxidase–conjugated or fluorophore (IRDye 800CW and IRDye 680RD)-conjugated, species-specific secondary antibodies. The fluorescence signals were captured and quantified using LiCOR Odyssey Imaging Systems.

*Sample preparation for proteomic analysis.* hCVF ($2 \times 10^6$) were seeded and grown to 80% confluence, and then treated with IL-1β (10 ng/mL) for 24 h. Cell culture media were collected and subjected to proteomic analysis. Conditioned media were concentrated using the ProteoSpin TM column (Norgen Biotek Corp, Canada). Concentrated samples were lysed with buffer (8 M urea, 1 mM sodium orthovanadate, 20 mM HEPES, 2.5 mM sodium pyrophosphate, 1 mM β-glycerophosphate, pH 8.0), followed by sonication (QSonica, LLC, Model no. Q55), and cleared by centrifugation[72]. The LC-MS/MS was performed on a fully automated proteomic technology platform that includes an Agilent 1200 Series Quaternary HPLC system (Agilent Technologies, Santa Clara, CA) connected to a Q Exactive Plus mass spectrometer (Thermo Fisher Scientific, Waltham, MA). The MS/MS spectra were acquired at a resolution of 17,500, with a targeted value of $2 \times 10^4$ ions or a maximum integration time of 200 ms. The ion selection abundance threshold was set at $8.0 \times 10^2$ with charge state exclusion of unassigned and z = 1, or 6–8 ions and dynamic exclusion time of 30 s. Peptide spectrum matching of MS/MS spectra of each file was searched against the human database (UniProt) using the Sequest algorithm within Proteome Discoverer v 2.3 software (Thermo Fisher Scientific, San Jose, CA).

*Cytokine and chemokine detection.* Cytokine and chemokine assays were performed by the Forsyth Multiplex Core (Cambridge, MA, USA) using the Human Cytokine/Chemokine Magnetic Bead Panel (Milliplex, Millipore Sigma, Burlington, MA, USA) and Bio-Plex®200 plate reader following manufacturers' specifications. Data were analyzed using Bio-Plex Manager software v6.0.

*Metabolomics.* Metabolomics of conditioned media was performed at the Beth Israel Deaconess Medical Center Mass Spectrometry Core according to published protocols[72]. hVCF ($2 \times 10^6$) were seeded and grown to confluence and treated with IL-1β (10 ng/mL) for 24 h. Conditioned media (1 mL) from the vehicle or IL-1β treated hVCF cells were used for the metabolomics analysis. Then, 500 µL of chilled −80 °C 80% methanol was added to 15 mL tubes containing 1 mL of conditioned media and evaporated using Speedvac to pellet the metabolites. SRM with polarity switching with a QTRAP 5500 mass spectrometer (AB/SCIEX) was used to assay 300 polar molecules[72].

*Illumina RNA Seq Analysis.* hVCF total RNA was extracted from flow-sorted Veh CD4+ Veh CD4− cells and IL-1β CD4+ and IL1β CD4− cells using Qiagen RNeasy Mini Kit (Cat# 74134) and the samples submitted to Genewiz for RNA Seq analysis. Initial sample QC analysis was performed followed by RNA library preparation with polyA selection and sequenced on Illumina Hi Seq in $2 \times 150$ bp paired end configuration. The raw data was obtained in FASTQ format and sequence reads were trimmed to remove possible adapter sequences and poor-quality nucleotides using Trimmomatic v.0.36. The trimmed reads were mapped to the Homo sapiens GRCh38 reference genome available on ENSEMBL using the STAR aligner v.2.5.2b. The unique gene hit counts were calculated using feature counts from Subread package v.1.5.2. And the gene hit counts were used for differential expression analysis. The gene expressions were compared using DESeq2 and the Wald test was used to generate the p values and log2 fold changes. The genes with an adjusted $P$-value < 0.05 and absolute log2 fold change >1 was considered as differentially expressed gene. Gene ontology on a unique set of genes for each population was performed using Database for Annotation, Visualization, and Integrated Discovery (DAVID) (https://david.ncifcrf.gov/) v6.8.

*Statistics and reproducibility.* The normal or Gaussian distribution for small sample sizes was determined using Shapiro-Wilk's test. For statistical comparisons between the two groups, 2-tailed unpaired *t*-test was used for unequal variances. When the sample sizes were small and not normally distributed, non-parametric tests 2-tailed Mann–Whitney U test was performed. Comparisons of >2 groups were made with 1- or 2-way ANOVA with either Tukey's post-hoc test as stated in the figure legends. Kruskal–Wallis tests followed by Dunn test comparison were used for continuous variables that did not show normal distribution. The values $*P < 0.01$ and $*P < 0.05$ were considered significant. The analyses were performed in GraphPad Prism version 8.0. Missing data are common in both proteomics and metabolomics due to heterogeneous responses to treatment or if the abundance of the protein or metabolite is below the detection limit of the mass spectrometer or a reduction in the electrospray performance. We used an approach that disregards variables containing missing data[65]. In some cases where there was a single missing value in one of the replicates, imputation using mean values was performed to replace the missing values. To control for multiplicity of the data, we employed the false discovery rate or Bonferroni correction.

**Reporting summary**. Further information on research design is available in the Nature Portfolio Reporting Summary linked to this article.

## Data availability

A preprint version of the manuscript is deposited on the bioRxiv site (https://doi.org/10.1101/2021.02.26.433023). The mass cytometry data is deposited in the flow Repository ID: FR-FCM-Z4P2 and the RNA seq data is deposited in GEO (GSE196195). The metabolomics data is deposited in the metabolomics workbench with datatrack_id:4171 and study_id:ST002805. The mass spectrometry proteomics data have been deposited to the ProteomeXchange Consortium via the PRIDE partner repository with the dataset identifier PXD044147. Source data for all figures are available in Supplementary Data 1.

## Code availability

We provide Supplementary code 1 (UECM Genes heatmap code), Supplementary code 2 (Expression from Genewiz2 code), and Supplementary code 3 (Inflammation gene heatmap code). Supplementary code 1 was used to create Supplementary Fig. 11f. Supplementary code 2 was used to create Fig. 6b as well as Supplementary Fig. 11b. Supplementary code 3 was used to create Supplementary Fig. 11e.

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

## Acknowledgements

We thank Dr. Ruth Montgomery, PhD, and Ms. Shelly Ren from the Yale CyTOF core facility and Dr. John M Asara, PhD, Director of the Mass Spectrometry core, Harvard Medical School, for helpful discussions. We thank Daniel Federick from Fluidigm for help with the tSNE analysis using CYTOBANK and Siraj Presswala for his assistance with the UMAP analysis using PYTHON and R tools. We also acknowledge Nagib Ahsan and Lelia Noble from the COBRE Center for Cancer Research Development, Proteomics Core Facility, Rhode Island Hospital, Providence, RI, for their help with proteomic sample processing and analysis. The authors also acknowledge support from Dr. Fenghai Duan of Advanced CTR (U54GM115677) for help with the statistical analysis of proteomics and metabolomics data. We thank Yang Zhou for his help with the Fulton index animal measurements. We thank the COBRE CPVB team at PVAMC for their help with mice

and rat echo's. The study was funded in part by grants from the National Institutes of Health P20 GM103652 Pilot Award (J.H.S.), National Institutes of Health RO1 HL130230 (S.R.), RO1 HL148598 and RO1 AI159078 (O.K.), Advance CTR 54GM115677 (J.H.S.), R56/R01 HL139680 (R.J.G., S.S.), R01 HL130356 (S.S.), R01 HL105826 (S.S.), R01 AR078001 (S.S.) R01 HL143490 (S.S.), Postdoctoral Fellowship HL125204 (M.J.I.), American Heart Association Postdoctoral Fellowship 830568 (M.J.I.), Burroughs Wellcome Fund fellowship BWF1022380 (M.J.I.), Rhode Island Foundation Grant (20190594) (J.H.S.) and TEAM UTRA grant from Brown University (J.H.S.). The metabolomic repository is supported by NIH grant U2C-DK119886 and OT2-OD030544 grants.

## Author contributions

Conceptualization: J.H.S., P.D., R.J.G.; Methodology: J.H.S., P.D., F.S.P., A.Z., M.J.I., J.P.G.-A., O.K., H.G., S.C., J.Y.; Investigation: J.H.S., F.S.P., A.Z., H.G., M.J.I., J.P.G.-A., S.C.; Visualization: J.H.S., R.J.G., M.J.I., J.P.G.-A., O.K., P.D. and S.R.; Funding acquisition: J.H.S., O.K., S.R.; Project administration: J.H.S.; Supervision: J.H.S., O.K., S.R., R.J.G.; Writing (original draft): J.H.S., R.J.G.; Writing (review; editing): J.H.S., R.J.G., O.K., P.D., S.R., J.Y., S.S.

## Competing interests

S.S. provided consulting and collaborative research studies to the Leducq Foundation (CURE-PLAN), Red Saree Inc, Greater Cincinnati Tamil Sangam, AavantiBio, Pfizer, Novo Nordisk, AstraZeneca, MyoKardia, Merck and Amgen. The other authors declare no competing interests.
