## [Peer Review File · Communications Biology]

Reviewers' comments:

Reviewer #1 (Remarks to the Author):

Authors for "IL-1 β -mediated adaptive re-programming of endogenous human cardiac fibroblasts to cells with immune features during fibrotic remodeling" unravels a novel observation that cardiac fibroblasts have the potential to differentiate to a type of fibroblasts expressing both alpha-SMA (myofibroblast marker) and CD4+ (a helper T-cell marker) under stimulation of IL-1beta. This conclusion was supported by well-designed experiments and data analysis. It may have a big impact on pathophysiology and pharmacology of fibrotic diseases.

1. TGF-beta promotes differentiation of fibroblasts to myofibroblasts and is involved in pathogenesis of fibrotic diseases. Nevertheless, IL-1beta and TGF-beta1 acts antagonistically in formation and differentiation of CD4 T cells. What is the role of TGF-beta in re-programing of cardiac fibroblasts to cells with immune features?
2. Is the conversion of cardiac fibroblasts into myofibroblasts with CD4 expression reversible or irreversible?
3. What are the levels of IL-1beta in circulation under basic or fibrotic remodeling?

Reviewer #2 (Remarks to the Author):

In this paper, the authors report that resident cardiac fibroblasts differentiate into CD4+ cells in response to injury or IL-1 β treatment. While the story seems very interesting, some key evidence is missing. For example, it is not clear if cells expressing both mesenchymal cell markers and T cell markers in Figure 2 were actually doublets. Also, the use of vimentin as the sole fibroblast marker in many experiments is not appropriate as it is expressed in most cell types. The purity of fibroblasts should be verified using antibodies targeting markers of other cell types, especially immune cell types, to eliminate possible contamination. Also, without proper lineage-tracing experiments, it is not possible to identify the source of cells expressing both fibroblast and T cell marker genes. Many figure panels were not mentioned in the text, making it hard to follow the paper.

1. Is there any relation between the top panel and bottom panel in Figures 1B and 1C? There is no red dotted line in Figure 1B.
2. PDGFR α is the gold standard marker for fibroblasts. The staining verifying the fibroblast identity in Figure S1 should include this marker.
3. Vim is an intermediate filament gene expressed in pretty much all cell types, including T cells (10.1172/JCI95713). It cannot be used as a fibroblast marker.
4. It is not clear how flow cytometry was done for Figure 1E.
5. Single-channel images are needed for immunostaining results.
6. Co-staining of aSMA and CD4 in Figure 2B needs to be done using fluorescence 2nd antibodies so that single-channel images can be provided, which is required to identify true double positive cells. In addition, aSMA is expressed in vascular smooth muscle cells (VSMCs) as well. Using aSMA alone as a myofibroblast marker has the risk of misidentifying VSMCs with a CD4+ cells located in the vessel as a double positive cell. Using PDGFR α as a fibroblast marker is more appropriate as it is also a cell membrane protein as CD4. Colocalization of the CD4 and PDGFR α will be a piece of much stronger evidence.
7. How were the results in Figure 3G generated? By imaging or flow cytometry? Again, SMA alone is not a good marker for fibroblasts or myofibroblasts.
8. What do the arrowheads in figure 3I point at?
9. Line 240. Trypan blue result was mentioned but not provided. Trypan blue stains cells that are already dead. TUNEL staining should be performed to rule out the possibility that the round cells are apoptotic cells. Also, some T cells may form doublets with fibroblast during cell isolation. It is known that some immune cells will express aSMA under certain circumstances. It is possible that the double positive cells are derived from T cells instead of fibroblasts. Fibroblast and T cell lineage-tracing mouse

lines are needed to generate a conclusion. Quantification for Figure 4E is needed. The experiments in Figure 4E should be done using sorted Pdgfra+ fibroblasts to eliminate all non-fibroblast cells.

Response to reviewers' comments

IL-1 β -mediated adaptive re-programming of endogenous human cardiac fibroblasts to cells with immune features during fibrotic remodeling

Reviewer #1 (Remarks to the Author):

Authors for "IL-1 β -mediated adaptive re-programming of endogenous human cardiac fibroblasts to cells with immune features during fibrotic remodeling" unravels a novel observation that cardiac fibroblasts have the potential to differentiate to a type of fibroblasts expressing both α SMA (myofibroblast marker) and CD4⁺ (helper T-cell marker) under stimulation of IL-1 β . This conclusion was supported by well-designed experiments and data analysis. It may have a big impact on pathophysiology and pharmacology of fibrotic diseases.

1. TGF- β promotes differentiation of fibroblasts to myofibroblasts and is involved in pathogenesis of fibrotic diseases. Nevertheless, IL-1 β and TGF- β 1 act antagonistically in formation and differentiation of CD4 T cells. What is the role of TGF-beta in reprogramming of cardiac fibroblasts to cells with immune features?

Ans: We thank the reviewer for posing this question and for insights on the role of other fibrotic and inflammatory mediators in cardiac fibroblasts trans-differentiation. TGF- β 1 is indeed a regulator of cell differentiation and phenotype switching and is implicated in many pro-inflammatory and reparative processes. In fact, TGF- β has been reported to promote activation and differentiation of lymphocyte by several investigators (Hofmann U, *Circulation* 2021, Wahl S.M, *J Leukoc Biol.*2004, and Oh Sa, *J Immunol.* 2013). TGF- β 1 is predominantly localized in the cardiomyocytes and the extracellular matrix, however its role in activating the cardiac fibroblasts is poorly defined. While we cannot currently rule out TGF- β mediated cardiac fibroblast differentiation, or the involvement of other cytokines, growth factors and pro-fibrotic mediators in transdifferentiation, we suggest that this can best be addressed in future work.

2. Is the conversion of cardiac fibroblasts into myofibroblasts with CD4 expression reversible or irreversible?

Ans: We thank the reviewer for this question related to possible bi-directionality of IL-1 β mediated cardiac trans-differentiation. Acknowledging its importance in the realm of disease pathogenesis and therapy we suggest that this question should best be considered in a future study. We speculate that such bi-directional control would be biologically beneficial as a certain percentage of resident cardiac fibroblasts would be needed to maintain homeostasis. Moreover, phenotypic changes demand elevated metabolic activity and biological resources, and therefore, energy conservation may play a role in restricting a permanent change in myofibroblast identity. We hypothesize that fibroblast plasticity constitutes a first line of defense mechanism which occurs before the arrival of exogenous macrophages and specialized immune cells and thereby comprises a reversible gain-of-function.

3. What are the levels of IL-1 β in circulation under basic or fibrotic remodeling?

Ans: We thank the reviewer for posing this important question related to physiological levels of IL1 β in fibrotic remodeling. We found increases of IL-1 β in the right ventricle of SUGEN hypoxia rat compared to the left ventricle of such animals (n=5, p<0.052). However, several cell types secrete IL-1 β apart from the macrophages and therefore, it might be challenging to predict in vivo cell specific secretion of IL-1 β under normal or pathological conditions. Moreover, PAH patient plasma levels indicate that IL-1 levels are associated with increased mortality. While systemic IL-1 levels are generally below detectable levels (Dinarello, *Blood*, 2011) in the general population, IL-1RA levels are easily detectable and have been used to determine IL-1 β levels. The threshold IL-1 level for promoting disease or affecting disease severity is unknown.

Reviewer #2 (Remarks to the Author):

In this paper, the authors report that resident cardiac fibroblasts differentiate into CD4⁺ cells in response to injury or IL-1 β treatment. While the story seems very interesting, some key evidence is missing. For example, it is not clear if cells expressing both mesenchymal cell markers and T cell markers in Figure 2 were actually doublets. Also, the use of vimentin as the sole fibroblast marker in many experiments is not appropriate as it is expressed in most cell types. The purity of fibroblasts should be verified using antibodies targeting markers of other cell types, especially immune cell types, to eliminate possible contamination. Also, without proper lineage-tracing experiments, it is not possible to identify

the source of cells expressing both fibroblast and T cell marker genes. Many figure panels were not mentioned in the text, making it hard to follow the paper.

1. Is there any relation between the top panel and bottom panel in Figures 1B and 1C? There is no red dotted line in Figure 1B.

Ans: We thank the reviewer for his comments. The dotted red line did not transfer in the final pdf. However, we have rectified our error in the revised version of the manuscript.

2. PDGFR α is the gold standard marker for fibroblasts. The staining verifying the fibroblast identity in Figure S1 should include this marker.

Ans: We do not completely concur that PDGFR α alone is the optimal marker of cardiac fibroblast identity and suggest that a combination of markers may be better used to confirm the fibroblast phenotype. Nonetheless, we have added an additional figure of primary human fibroblast positive for PDGFR α in addition to other known fibroblast markers.

3. Vim is an intermediate filament gene expressed in pretty much all cell types, including T cells (10.1172/JCI95713). It cannot be used as a fibroblast marker.

Ans: We agree with the reviewer that most cell types express Vimentin. However, we also note that Vimentin is commonly used to differentiate resident cardiac fibroblasts from activated cardiac myofibroblasts. In our work, we have not determined the phenotype such cells solely on the basis of Vimentin expression, but with a combination of imaging, flow cytometry, RNA sequencing, cell phenotyping, differentiation assays, proteomics, metabolomics and functional analyses of secreted cytokines and chemokines, as well as *in vivo* lineage tracing.

4. It is not clear how flow cytometry was done for Figure 1E.

Ans: We thank the reviewer for these comments. Figure 1E is the 2D projection of the cell expression markers determined using mass cytometry that uses multiple heavy metal tagged antibodies and produces fcs files that can be re-analyzed using floJo or other FACS software. The histogram in Figure 1E compares quantitatively the expression of CD4 expressing cells in different cell phenotypes (SMA+ Vim-, SMA-Vim+ and SMA+Vim+). We have now included this information in the figure legends and more detailed information in the materials and methods section.

5. Single-channel images are needed for immunostaining results.

Ans: We thank the reviewer for these comments. We have provided single channel images in supplementary data due to limitations in Figure 1 that is crowded with information.

6. Co-staining of α SMA and CD4 in Figure 2B needs to be done using fluorescence 2nd antibodies so that single-channel images can be provided, which is required to identify true double positive cells. In addition, α SMA is expressed in vascular smooth muscle cells (VSMCs) as well. Using α SMA alone as a myofibroblast marker has the risk of misidentifying VSMCs with CD4+ cells located in the vessel as a double positive cell. Using PDGFR α as a fibroblast marker is more appropriate as it is also a cell membrane protein as CD4. Co-localization of the CD4 and PDGFR α will be a piece of much stronger evidence.

Ans: The double immunohistochemistry was performed on FFPE human tissues stored for an unknown length of time. Apart from this being a tricky method to retrieve antigen, it is almost impossible to perform dual immunofluorescence staining on FFPE tissues without considerable background and without suitable antibodies for fluorescence staining. We show using flow cytometry that human cardiac fibroblasts also express PDGFR α and CD4 in addition to α SMA. In *pdgfra* specific transgenic mice, 92% of the cells lose the *pdgfra* marker upon induction with IL-1 β . Panel 7H and 7I suggest that with differentiation, the phenotypic characteristics of myofibroblasts are altered.

7. How were the results in Figure 3G generated? By imaging or flow cytometry? Again, SMA alone is not a good marker for fibroblasts or myofibroblasts.

Ans: Figure 3G is the quantification of the CD3+ CD4+ cells counted using flow cytometry in both Normoxia and Sugen Hypoxia rats. Figure 3H is the quantification of α SMA+CD4+ immunofluorescence images. We have rectified the confusion by placing arrows to explain this phenomenon better in the figures.

8. What do the arrowheads in figure 3I point at?

Ans: We thank the reviewer for helping us improve the details of the figure. We apologize for the missing information on the yellow arrows in the figures. We have now indicated what the yellow arrows are pointing at in the figure legends. For your reference, the yellow arrow is pointing at the cells expressing α SMA or CD4 markers.

9. Line 240. Trypan blue result was mentioned but not provided. Trypan blue stains cells that are already dead. TUNEL staining should be performed to rule out the possibility that the round cells are apoptotic cells. Also, some T cells may form doublets with fibroblast during cell isolation. It is known that some immune cells will express α SMA under certain circumstances. It is possible that the double positive cells are derived from T cells instead of fibroblasts. Fibroblast and T cell lineage-tracing mouse lines are needed to generate a conclusion. Quantification for Figure 4E is needed. The experiments in Figure 4E should be done using sorted Pdgfra+ fibroblasts to eliminate all non-fibroblast cells.

Ans (abridged): Since the cells were live cells and colorless, it was not possible to record any dead cells in this setting. We also note that no apoptotic markers show up in the RNA Seq of CD4+ α SMA+ cells. Moreover, there is strong evidence that these cells show secretory characteristics that would diminish the necessity of a TUNEL assay (cell death). We are thankful to the reviewer for suggesting the bi-directionality of the process from T-cells to fibroblast, but as noted above, this is outside of our current focus.

We have provided evidence from lineage traced mouse lines that are periostin positive or tcf-21+ that demonstrate the acquisition of immune cell markers and loss of *pdgfra* marker by 92% of the endogenously labelled cardiac fibroblast cells in the presence of IL-1 β (Figure 7). Flow cytometry experiments were performed to quantify *pdgfra* fibroblast population, as suggested by the reviewer (Figure 7 and Figure S14).

Reviewers' comments:

Reviewer #1 (Remarks to the Author):

This is a revised manuscript. Authors answered questions and provided explanations to some concerns. There are some new questions related to experimental design of this manuscript.

1. In Cardiac fibroblast culture in ImmunoCult™ –XF T-Cell Expansion Media (p. 23), IL-1beta at a concentration of 10 ng/mL was added to hVCF (?) for 96 hours. Please provide rationale for selection of the time and concentration of IL-1beta.
2. CD3/CD28 T-Cell Activator was added after 72 hours of culture. Cells were cultured in T-cell expansion media with T Cell Activator in the presence or absence of IL-1 β for 14 days (p. 24). Please provided name and resources of vendor for T cell activator. In Fig. 6 for cell differentiation of hCVF, IL-1beta was added for 10 days. Please explain the discrepancy of this time point.
3. For proteomic and metabolomics, IL-1beta (10 ng/mL) was added for 24 hours. Multiple time points of incubation are recommended.

Reviewer #2 (Remarks to the Author):

1. The author indicated that "we have added an additional figure of primary human fibroblast positive for PDGFR α in addition to other known fibroblast markers". However, no relevant data was found.
 2. Figure 7e, percentages of CD45+ and CD4+ cells in vehicle groups are also needed.
 3. Figures 7f and 7h are possibly mislabeled. There is almost no cell over 104 in the IL1b group in CD4 FACS graph shown in Figure 7e. But a major peak is seen in figure 7h. Can the authors explain why? Also, there is no apparent difference between vehicle and IL1b in CD4 (Figure 7e). The IL1b simply has more cells shown in the graph.
 4. Y axis in Figure 7e should be replaced with tdtomato. And IHC and ICC show tomato+;CD4+ cells are needed.
 5. "a sub-population of endogenous human cardiac fibroblasts expressing increased levels of CD4+, a helper T-cell marker". CD4 or CD4+?
 6. A more detailed description of the CyTOF cell labeling procedure is needed. Reference 47 was cited but did not include a detailed description of the labeling procedure. How was the simultaneous staining of intracellular and cell surface markers achieved?
 7. "CD3+CD4- cardiac fibroblast populations were elevated in RV under both Nx (16.39% \pm 14.06%) and SuHx conditions (17.24% \pm 15.46%) at comparable levels and were found to be less prominent in LV, under both Nx (2.10% \pm 0.71%) and SuHx conditions (9.41% \pm 6.04%) (Figure 3, Panel G)." Elevated compared to what?
 8. "Percentages of CD3+CD4+ populations expressing cardiac fibroblast cells showed some increase in LV (5.79% \pm 4.92%) under SuHx conditions in comparison to Nx, (5.50% \pm 5.75%) although it did not reach significance (Figure 3, Panel G)." where is the data?
- There are many other errors in the text and figure legends which need to be addressed.

Reviewers' comments:

IL-1 β -mediated adaptive re-programming of endogenous human cardiac fibroblasts to cells with immune features during fibrotic remodeling

Reviewer #1 (Remarks to the Author):

This is a revised manuscript. Authors answered questions and provided explanations to some concerns. There are some new questions related to the experimental design of this manuscript.

1. *In cardiac fibroblast culture in ImmunoCult™ –XF T-Cell Expansion Media (p. 23), IL-1 β at a concentration of 10 ng/mL was added to hVCF (?) for 96 hours. Please provide rationale for selection of the time and concentration of IL-1 β .*

The concentration of IL-1 β used is based on multiple previous studies (Kim EH, FASEB, 2018; Lee JH, JACS, 2010; Adelus ML, Circ, 2022) as well as our own dose and time dependent proliferation experiments using primary human cardiac fibroblast cells and rat cardiac fibroblast cells. Through these experiments, we observed with bright field microscopy that the first change in cell morphology occurred at 96 hours. Therefore, this time point was used for all subsequent experiments. We have included this information in the results section of the manuscript (pp, 9, lines 1-3).

2. *CD3/CD28 T-Cell Activator was added after 72 hours of culture. Cells were cultured in T-cell expansion media with T Cell Activator in the presence or absence of IL-1 β for 14 days (p. 24). Please provide the name and resources of vendor for T cell activator. In Fig. 6 for cell differentiation of hCVF, IL-1 β was added for 10 days. Please explain the discrepancy of this time point.*

In addition to the catalog number that was already provided (Methods), we have now included the vendor name and location on pp 23, last paragraph.

3. *For proteomics and metabolomics, IL-1 β (10 ng/mL) was added for 24 hours. Multiple time points of incubation are recommended.*

We thank the reviewer for pointing out the importance of this variable. This duration of incubation that we represent is the minimum time needed to observe a significant biological effect. We have clarified this point in the revised manuscript. We agree fully with the reviewer that longer time points would produce valuable data in regard to the complexity of the inflammo-fibrotic response mediated by IL1- β in these (and similar) cell types. These experiments, while of certain value, are, however, beyond the focus of the current paper.

Reviewer #2 (Remarks to the Author):

1. *The author indicated that “we have added an additional figure of primary human fibroblast positive for PDGFR α in addition to other known fibroblast markers”. However, no relevant data was found.*

We apologize for not clarifying. This figure is presented in the supplementary data Supp Figure S14 (see below). We demonstrate that human fibroblasts which start as PDGFR α acquire CD4 positivity.

Figure S14

a Mouse Pdgfr α + cardiac fibroblasts are neither CD4 nor CD45 + at baseline

b Human cells that start as Pdgfra acquire CD4

2. *Figure 7e, percentages of CD45+ and CD4+ cells in vehicle groups are also needed.*

We have updated the figure with the percentages for the vector controls.

3. *Figures 7f and 7h are possibly mislabeled. There is almost no cell over 10⁴ in the IL1b group in CD4 FACS graph shown in Figure 7e. But a major peak is seen in figure 7h. Can the authors explain why? Also, there is no apparent difference between vehicle and IL1b in CD4 (Figure 7e). The IL1b simply has more cells shown in the graph.*

We thank the reviewers for careful examination of this figure. We have included the vehicle percentages in Figure 7e. Due to the lack of robust shift in the mouse cells with the use of CD4 antibody we have analyzed the data by normalizing the signal and made sure the shift we observed is representation of CD4+ identity. The peak, therefore, observed at the 10⁴ is not a direct representation of the FACS plots. The majority of the negative cells in FACS plots (the red core in Figure 7e CD4 plots) are just past the 10² marks whereas in the normalized data the major peak for the vehicle is now between 10³ and 10⁴ (in figure 7h) which shows that we treated both samples equally for this postprocessing. We also showed the data in Figure 7i as bar graphs including the human data shown in the supplement which shows a more robust separation in FACS plots due to better antibody affinity.

4. *Y axis in Figure 7e should be replaced with tdTomato. And IHC and ICC show tomato+;CD4+ cells are needed.*

We appreciate the suggestion. We acknowledge in our data presentation that majority of the fibroblast did not convert to CD45/CD4 cells (Figure 7g) however the ones that converted lost their pdgfra or mefsk4 expression (Figure 7f). Therefore, we did not want to reduce impact by showing that the majority of tdTomato cells remain as fibroblast in Figure 7e. However, to discuss this point we have shown this data in Figure 7g where the black line histogram shows that majority of these tdTomato cells are not CD45+. We believe this presentation highlights the significance of the fate change but also accurately depicts the frequency of this phenomenon.

5. *“a sub-population of endogenous human cardiac fibroblasts expressing increased levels of CD4+, a helper T-cell marker”. CD4 or CD4+?*

Thank you for the correction. This should be simply “CD4”.

6. *A more detailed description of the CyTOF cell labeling procedure is needed. Reference 47 was cited but did not include a detailed description of the labeling procedure. How was the simultaneous staining of intracellular and cell surface markers achieved?*

The details of the cell labeling procedure are included on pp 22 under cell labeling methods. The staining is similar to conventional immunofluorescence labelling except for the use of metal tag antibodies. We performed mostly membrane marker staining and not intracellular staining as most of the immune cells had surface markers. We have generally included more detail related to the cell labeling procedure.

7. *“CD3+CD4- cardiac fibroblast populations were elevated in RV under both Nx (16.39% ± 14.06%) and SuHx conditions (17.24% ± 15.46%) at comparable levels and were found to be less prominent in LV, under both Nx (2.10% ± 0.71%) and SuHx conditions (9.41% ± 6.04%) (Figure 3, Panel G).” Elevated compared to what?*

We thank the reviewer for improving the content of this section. This experiment is a detailed characterization of CD3 and CD4 and CD3CD4 populations in both right and left ventricles. We have revised the text to explain this further on pp 6.

8. *“Percentages of CD3+CD4+ populations expressing cardiac fibroblast cells showed some increase in LV (5.79% ± 4.92%) under SuHx conditions in comparison to Nx, (5.50% ± 5.75%) although it did not reach significance (Figure 3, Panel H).” Where is the data?*

There is no mention of Figure 3, Panel H in the text.

Additional comment: There are many other errors in the text and figure legends which need to be addressed.

We have performed an additional check of the text for errors and made all corrections.

REVIEWERS' COMMENTS:

Reviewer #2 (Remarks to the Author):

The reviewers addressed most of my comments. However, the response to my comment #4 is not justified. The use of *Pdgfra* lineage tracing mice in Figure 7 will ensure that all fibroblasts, regardless of the expression of CD4/CD45, permanently express tdTomato. Changing the Y axis in Figure 7e will not affect the cells included in the panel but allow the verification of tdTomato expression in CD4 and CD45 expressing fibroblasts.